# Genetic influences on brain and cognitive health and their interactions with cardiovascular conditions and depression

Peter Zhukovsky [1,2,3], Earvin S. Tio [3,4], Gillian Coughlan[5], David A. Bennett[6], Yanling Wang [6], Timothy J. Hohman [7,8], Diego A. Pizzagalli [9], Benoit H. Mulsant [1,2], Aristotle N. Voineskos [1,2,12] ✉ & Daniel Felsky [2,3,4,10,11,12] ✉

Approximately 40% of dementia cases could be prevented or delayed by modifiable risk factors related to lifestyle and environment. These risk factors, such as depression and vascular disease, do not affect all individuals in the same way, likely due to inter-individual differences in genetics. However, the precise nature of how genetic risk profiles interact with modifiable risk factors to affect brain health is poorly understood. Here we combine multiple data resources, including genotyping and *postmortem* gene expression, to map the genetic landscape of brain structure and identify 367 loci associated with cortical thickness and 13 loci associated with white matter hyperintensities ($P < 5 \times 10^{-8}$), with several loci also showing a significant association with cognitive function. We show that among 220 unique genetic loci associated with cortical thickness in our genome-wide association studies (GWAS), 95 also showed evidence of interaction with depression or cardiovascular conditions. Polygenic risk scores based on our GWAS of inferior frontal thickness also interacted with hypertension in predicting executive function in the Canadian Longitudinal Study on Aging. These findings advance our understanding of the genetic underpinning of brain structure and show that genetic risk for brain and cognitive health is in part moderated by treatable mid-life factors.

Evidence suggests that modifying certain health-related factors in mid-late life can improve risk trajectories for late-life dementia[1]. Specifically, treating hypertension, depression, hearing impairment, smoking, obesity, or diabetes; reducing excessive alcohol consumption; and sustaining physical activity and social contact is beneficial. Several of these risk factors, including major depression[2,3] and cardiovascular disease[4,5], are also associated with differences in brain structure and may influence the risk for dementia via this mechanism[6,7]. Similarly, hypertension is

[1]Campbell Family Mental Health Research Institute, Centre for Addiction and Mental Health, Toronto, ON M5T 1R8, Canada. [2]Department of Psychiatry, Temerty Faculty of Medicine, University of Toronto, Toronto, ON M5T 1R8, Canada. [3]Krembil Centre for Neuroinformatics, Centre for Addiction and Mental Health, Toronto, ON, Canada. [4]Institute of Medical Science, Temerty Faculty of Medicine, University of Toronto, Toronto, ON M5S 1A8, Canada. [5]Department of Neurology, Massachusetts General Hospital, Boston, MA 02129, USA. [6]Department of Neurological Sciences, RUSH Medical College, Chicago, IL 60612, USA. [7]Vanderbilt Memory & Alzheimer's Center, Vanderbilt University Medical Center, Nashville, TN 37232, USA. [8]Vanderbilt Genetics Institute, Vanderbilt University Medical Center, Nashville, TN 37232, USA. [9]Department of Psychiatry, Harvard Medical School and Center for Depression, Anxiety and Stress Research, McLean Hospital, Belmont, MA 02478, USA. [10]Dalla Lana School of Public Health, University of Toronto, Toronto, ON M5S 1A8, Canada. [11]Rotman Research Institute, Baycrest Hospital, Toronto, ON M6A 2E1, Canada. [12]These authors contributed equally: Aristotle N. Voineskos, Daniel Felsky. ✉e-mail: aristotle.voineskos@camh.ca; daniel.felsky@camh.ca

associated with a greater risk of severe white matter lesions, which in turn is associated with dementia risk[8,9]. However, brain structure is highly heritable[10], and it remains unclear whether the effects of treatable conditions on cortical thinning and cerebrovascular disease are different in individuals with different genetic risk profiles. Identifying such gene-by-environment interactions is, therefore, an essential step toward developing precision interventions. They can inform whether modifying dementia risk factors in mid-life should be expected to succeed despite the presence of genetic risk for cortical thinning or white matter lesions.

Cortical thinning is a hallmark of neurodegenerative diseases[11–13] and is the earliest established and most widely replicated biomarker of Alzheimer's disease severity[14–17]. Unlike surface area and sulcal depth, which are more heritable[18], cortical thickness is under relatively greater environmental influence[19]. Cortical thickness declines most rapidly in mid- and late-life[20]. White matter lesions, quantified as white matter hyperintensities (WMH) on MRI scans, are a critical marker of cerebral small vessel disease and also contribute to risk for dementia, cognitive impairment[21], and disability[22,23].

Previous large-scale genome-wide studies of younger adults with psychiatric conditions in the ENIGMA cohort[24] and the lifespan CHARGE cohort[25] have identified common genetic variants that influence cortical thickness. A larger number of genomic risk loci for cortical thickness have also been identified in a recent analysis of 3144 brain imaging phenotypes in older adults from the UK Biobank (UKB)[26], yet these loci have not been explored in detail. Similarly, recent large-scale analyses of WMH have uncovered a number of independent risk loci[27]. While the genetic control over these heritable and clinically important brain phenotypes has been explored, the interactions between modifiable factors and genetics are not known.

In this context, we use the large-scale mid- and late-life UKB cohort to further explore the genetics of cortical thickness and WMH. Second, we performed cross-region genetic correlation analyses and triangulated evidence for functionally significant genes using RNA sequencing and matched MRI in an independent late-life sample from the Religious Orders Study (ROS) and Memory and Aging Project (MAP)[28]. We hypothesized that a subset of genetic variants associated with cortical thickness and WMH would create a disproportionate risk for cortical thinning in the presence of major depression or cerebrovascular disease, detectable as gene-by-environment interactions. Finally, we explored the impact of identified variants on memory and executive function in a third population-based mid- and late-life cohort, the large-scale Canadian Longitudinal Study on Aging (CLSA)[29–31].

## Results

### GWAS of cortical thickness and WMH

Building on existing GWAS of cortical thickness and WMH, we analyzed the UKB MRI cohort. We performed GWAS in two stages: (1) an omnibus test on global thickness ($n = 34,552$) and total WMH volume ($n = 30,708$), and (2) a set of 33 independent GWAS on regional thickness of 33 cortical regions. Unlike most existing GWAS of cortical thickness[18,24,25], in this study, we co-varied for total intracranial volume instead of global thickness, similar to neuroimaging studies of psychiatric disorders[32–35].

Our omnibus GWAS identified 22 independent loci (at uncorrected $p < 5.0 \times 10^{-8}$) associated with global thickness and 13 loci associated with total WMH volume (Fig. 1, Supplementary Data 3 and 4). Among the 13 loci associated with WMH, 10 have been

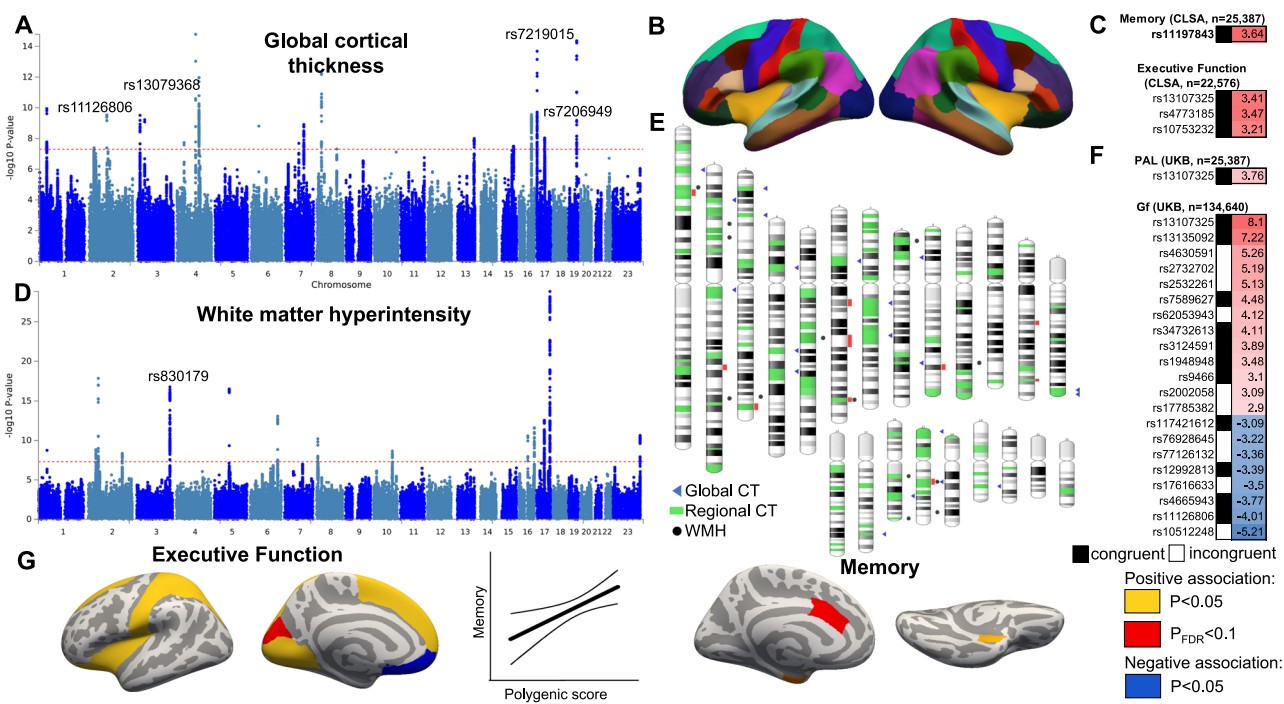

**Fig. 1 | Locus discovery for global cortical thickness and white matter hyperintensities. A**, **D** Manhattan plots of loci associated with global cortical thickness (CT) and white matter hyperintensities (WMH) show the $-\log_{10}(p\text{-value})$ for each genomic location. A red line denotes the genome-wide significance threshold of two-sided $P < 5 \times 10^{-8}$. Results after correction for the 34 cortical thickness GWAS are shown in Supplementary Data 4. **B** Desikan–Killiany parcellation was used for regional thickness analyses (33 bilateral regions). **E** Ideogram of loci that influence global and regional cortical thickness and WMH. GWAS analyses were conducted in the UK Biobank ($n = 35,846$). T-statistics for genetic variants associated with memory and executive function in the Canadian Longitudinal Study of Ageing

(**C**, CLSA) and in the UKB (**F**) after false discovery rate correction ($q < 0.1$). Congruent effects on cortical thickness and cognition are highlighted with a black square. **G** Polygenic scores for the thickness of prefrontal regions, such as the insula or superior frontal gyrus, were associated with executive function, while polygenic scores for the thickness of the anterior cingulate and entorhinal cortices were associated with memory in CLSA. Regions with a positive association at uncorrected $p < 0.05$ are highlighted in yellow; regions with a negative association at uncorrected $p < 0.05$ are highlighted in blue, while regions with a positive association at $p_{FDR} < 0.1$ are highlighted in red. PAL paired associates learning, Gf fluid intelligence.

previously associated with WMH[27], either as an association with the exact same variant or a variant in moderate-strong LD ($r^2 > 0.4$). Among the 22 loci significantly associated with global thickness, only five were in high LD ($r^2 > 0.4$) with the genome-wide significant loci from the most recent ENIGMA GWAS[24], likely since we did not include global thickness as a covariate and due to the more homogeneous makeup of our UKB sample compared to ENIGMA.

In 33 regional GWAS analyses, we identified an additional 345 risk loci for cortical thickness of 33 cortical regions (at uncorrected $p < 5.0 \times 10^{-8}$) and provided clumped results across all cortical thickness GWAS in Supplementary Data 5. A large proportion (59.4%) of independently significant SNPs for global thickness in our UKB analyses were also significant at a suggestive significance threshold ($p < 5 \times 10^{-4}$) in ENIGMA. Further, 21 associations from the discovery analysis also passed a liberal replication threshold ($p < 0.05$) in a cis-replication analysis of the ROS/MAP cohort.

### Regional heritability, genetic correlations, and gene ontology

To identify areas of the brain with shared and distinct genetic control over cortical aging, we estimated the heritable components of cortical thickness and WMH phenotypes and between-region genetic correlations for cortical thickness. SNP-based heritability estimates were $h^2_{SNP} = 0.25$ (SE = 0.021) for global thickness and $h^2_{SNP} = 0.22$ (SE = 0.047) for WMH. Compared to published GWAS of thickness and WMH, we observed the expected strong genetic correlations between our results and those from the ENIGMA consortium's global thickness analysis ($r_g = 0.82$, $p = 5 \times 10^{-95}$) and from Persyn et al.'s analysis of WMH[27] ($r_g = 0.976$, $p = 6 \times 10^{-70}$). We also tested for genetic correlations of global thickness and WMH with a set of psychiatric and neurological traits (Fig. 2C). While WMH showed significant genetic correlations with insomnia, ADHD, and intelligence (Supplementary Data 17), no significant genetic correlations were found for global thickness after multiple comparison correction.

Across regions, the heritability of cortical thickness varied substantially, with pre- and postcentral regions, the precuneus, and visual cortex regions showing the highest heritability of 0.23-0.28, while

entorhinal and anterior prefrontal regions showed the lowest heritability ($h^2_{SNP} = 0.1$, Supplementary Data 16). Regional genetic correlations were generally high, consistent with previous work[24,25], though we noted comparatively low genetic correlations between parahippocampal, entorhinal, and caudal anterior cingulate regions with all other cortical regions (Fig. 2B). A network modularity analysis revealed three distinct modules (Fig. 2A): first, a posterior module including visual, motor and temporal areas, second, a midline module centered around cingulate areas that also included the insula, and medial orbitofrontal cortex and third, a prefrontal module. Notable outliers were the parahippocampal cortex in the midline module and the entorhinal cortex in the prefrontal module, which overall showed low heritability and genetic correlations with other regions. GO enrichment analyses implicated terms describing biological processes, including cell development and growth (Supplementary Fig. 5).

### Overlap in genes identified by GWAS eQTL mapping and postmortem brain gene expression analysis

Following our exploration of the genetic control over cortical thickness and WMH, we performed a direct analysis of postmortem differential gene expression in the independent ROS/MAP sample of the frontal cortex with matched *antemortem* MRI ($n = 66$). This analysis revealed 223 genes significantly associated with caudal middle frontal thickness ($p_{FDR} < 0.05$, Supplementary Data 24, 25). In parallel, we performed eQTL mapping of genetic loci from our GWAS using eQTL data from GTEx (using frontal cortex and BA9 references in FUMA). Combining these results, we tested the overlap in the direction of expression effects for all genes passing quality control in both analyses. Overall, we found strong agreement in the direction of effect between GWAS and differential expression results for global thickness (77% of associations were in the same direction; Fisher's $p = 0.04$) and rostral middle frontal thickness (73%, Fisher's $p = 0.03$) but not for caudal middle frontal thickness (45%, Supplementary Data 9; Fisher's $p = 0.99$).

Among genes with concordant effects in both analyses, *STMN4*, a stathmin-coding gene primarily expressed in neurons[36], despite its

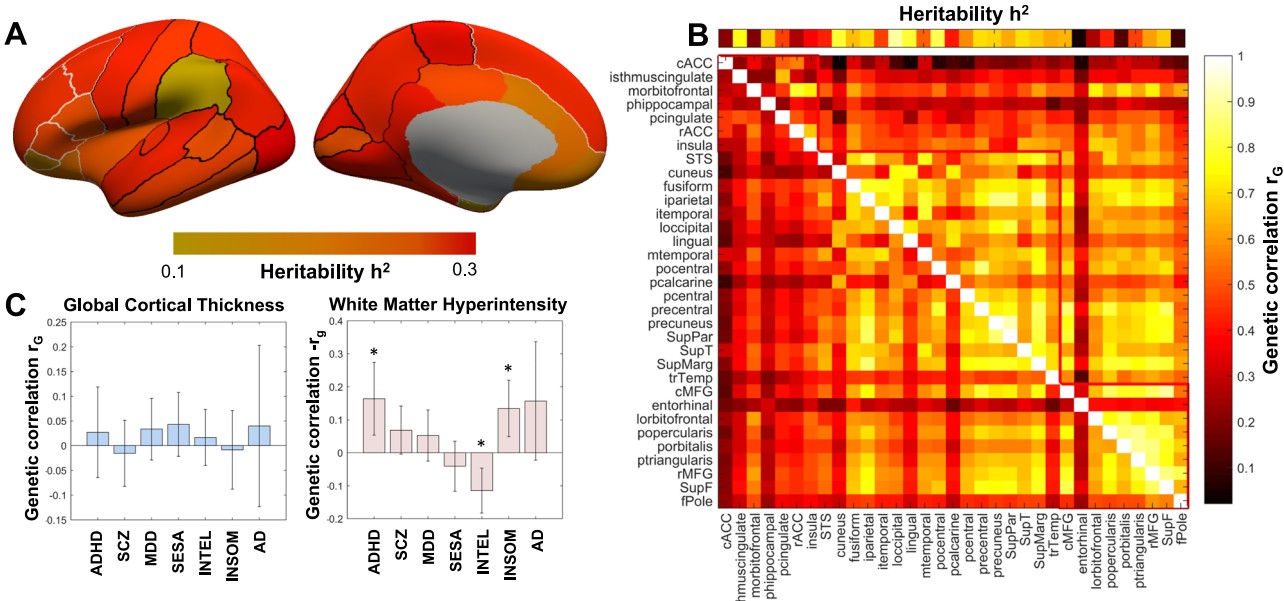

**Fig. 2 | Heritability and genetic correlations. A** Heritability estimated using UKB GWAS results ($n > 34,500$) was lowest for the entorhinal, frontal pole, anterior cingulate, and orbitofrontal cortices. **B** Genetic correlations (from LD Score) between cortical regions without covarying for global cortical thickness are shown above the diagonal, while phenotypic correlations between regional cortical thickness pairs are shown below the diagonal. **C** Genetic correlations between

global thickness and white matter hyperintensity with attention deficit and hyperactivity disorder (ADHD), schizophrenia (SCZ), major depressive disorder (MDD), sensitivity to environmental stress and adversity (SESA), intelligence (INTEL), insomnia (INSOM) and Alzheimer's disease (AD). Error bars represent 95% confidence intervals. *two-sided Bonferroni corrected $p < 0.007$ (0.05/7).

lower RNA abundance, was significantly associated with increased caudal middle frontal thickness in the postmortem analysis ($t = 3.74$, $p = 0.00047$, $p_{FDR} = 0.04$) and to global thickness in the UKB GWAS eQTL mapping analysis ($p = 1.74 \times 10^{-13}$, $Q_{FDR} = 7.16 \times 10^{-9}$). Six other genes (*DPYSL5, KANSL1, ARL17A, ARL17B, LRRC37A, LRRC37A2*) that were identified through UKB GWAS eQTL analyses also passed a more liberal, uncorrected threshold ($p_{UNCORRECTED} < 0.05$) in the postmortem analysis, although the direction of the association was inconsistent for *KANSL1* and *DPYSL5*. Similarly, five genes (*STMN4, TRIM35, CLEC18C, ARHGAP27, ARL17A*) were significantly associated with rostral middle frontal thickness both in the postmortem expression analysis and in the GWAS eQTL mapping, with a consistent direction of the effect for all genes except *CLEC18C* (Supplementary Data 8).

### Convergent evidence for a complex locus on Chromosome 17 influencing thickness of multiple brain regions

Among hundreds of loci exerting effects on imaging phenotypes, we noted that the complex q21.31 region of chromosome 17 harbored several variants and genes of functional significance to global and regional cortical thickness (Fig. 3). This region included two independent SNPs associated with thickness (rs7206949 and rs55684829), and a high number of significant SNPs in LD over a ~1 MB region encompassing over a dozen genes. eQTL mapping using expression data from the frontal cortex and BA9 linked these SNPs to 25 genes, and our postmortem RNAseq findings show that *LRRC37A, LRRC37A2*, and *ARL17A, ARL17B* play a functional role, as their expression was associated with the caudal middle frontal thickness of the PFC in the independent ROS/MAP cohort.

Using the SEA-AD Brain Cell Atlas, which includes single-nucleus RNAseq data from middle temporal gyrus of 84 elderly donors (42 cognitively normal and 42 with dementia), we found that these genes were preferentially expressed in microglia, astrocytes, oligodendrocytes, and oligodendrocyte precursor cells alongside vascular and leptomeningeal cells (Fig. 3E, Supplementary Fig. 1). In addition to these genetic variants linked to global thickness, we also found a number of genomic risk loci related specifically to 13 visual, temporal,

parietal, and prefrontal regions. The anatomical anterior-posterior ordering of these regions mirrored the order of their associated SNP location on q21.31, i.e., SNPs associated with the thickness of prefrontal regions were closer to the centromere than those associated with visual and temporal regions, except for the lateral occipital cortex (Supplementary Data 23). To disentangle the associations between multiple genetic variants and multiple phenotypes, we used partial least squares regression (Supplementary Section 7). We show that while some SNPs were uniquely linked to the cortical thickness of specific regions, we also found multivariate patterns of association, as several SNPs were linked to the cortical thickness of several regions.

Given the presence of a large *MAPT* haplotype at 17q21[37], we also report the associations of haplotype tagging SNPs[38] with global cortical thickness and LD between these SNPs with loci we show in Fig. 3A. We found signals for cortical thickness both within and outside of the *MAPT* haplotype. Briefly, a tagging SNP for the protective H2 haplotype (rs8070723) was in high LD with several SNPs identified in our GWAS (Supplementary Fig. 6) and was associated with global ($p = 4 \times 10^{-6}$) and regional thickness (Supplementary Data 19). However, our GWAS also identified other SNPs from this region that were not in LD with the *MAPT* haplotype.

### Influences of modifiable risk factors for dementia and gene-by-environment interactions

Following our mapping of the genetic drivers of volume-corrected cortical thickness and WMH in UKB, we similarly mapped the associations of depression and cardiovascular disease on the same phenotypes. We have also previously shown, a small but robust effect of major depression on cortical thinning[2]. The presence of cardiovascular conditions was associated with significant differences in thickness for nearly half of all regions tested ($p_{FDR} < 0.05$), with the strongest cortical thinning observed in insular and opercular regions (Fig. 4A).

Combining these risk factors with our top newly identified genetic predictors of cortical thickness, we tested for GxE interactive effects on regional thickness. We prioritized 220 unique variants we identified in the GWAS for these analyses. For each genetic variant, we first identified all brain regions with evidence for

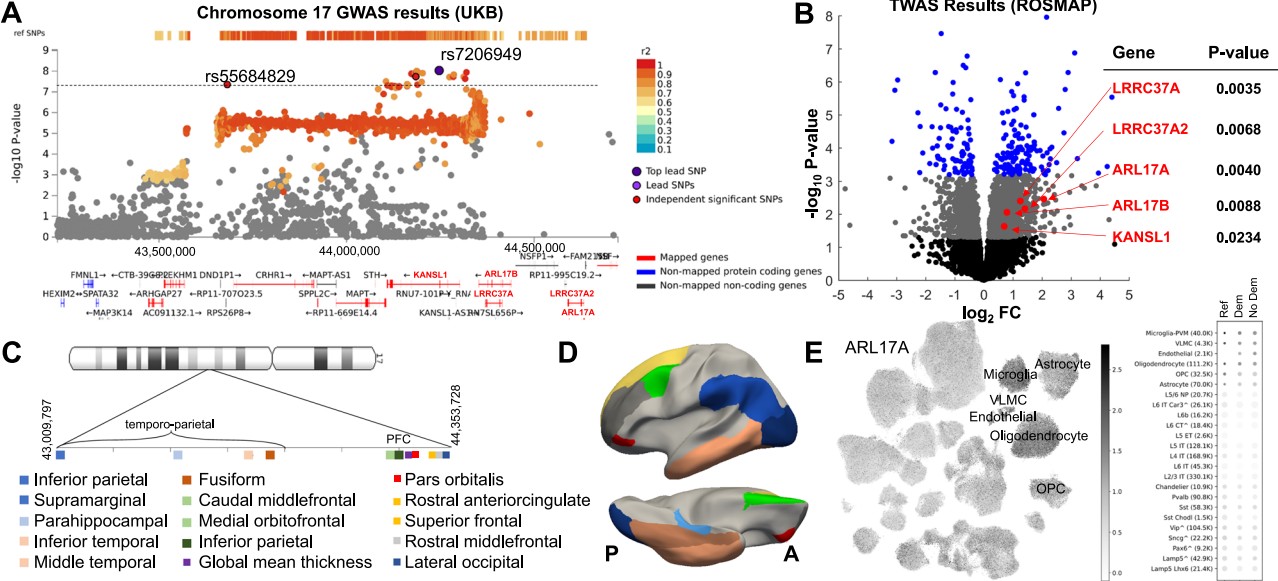

**Fig. 3 | Complex locus influencing cortical thickness on Chromosome 17.**
**A** Regional plot for rs7206949, including additional variants in high linkage disequilibrium and anatomically mapped genes (UKB, *n* > 34,500). **B** Volcano plot of the postmortem differential gene expression analysis results from ROS/MAP (*n* = 66). **C** Ideogram of 17q21.31 that includes the significant genomic risk loci and associated cortical regions. **D** Anterior−posterior organization of the cortical

regions and the genetic variants associated with these cortical regions. **E** Seattle Alzheimer's Disease Brain Cell Atlas map of tissue-specific expression values for the *ARL17A* gene, showing increased expression in microglia, endothelial cells, astrocytes, and oligodendrocytes. PFC prefrontal cortex, OPC oligodendrocyte precursor cells, VLMC vascular and leptomeningeal cells, FC: fold change.

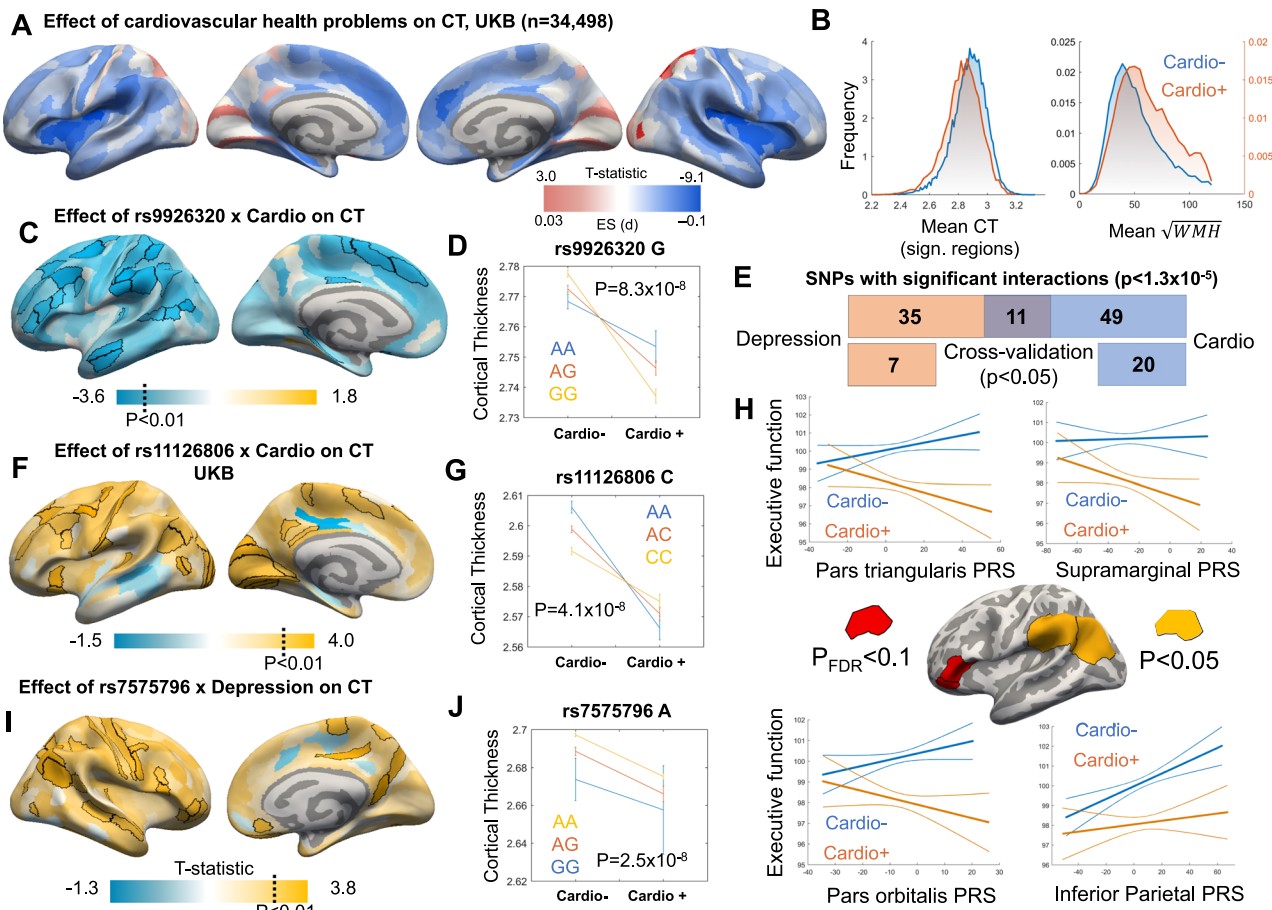

**Fig. 4 | Gene-by-environment interactions on cortical thickness.**
**A**, **B** Cardiovascular conditions reduced cortical thickness (CT) in the insular regions and increased the total volume of white matter hyperintensities (WMH). Examples of interactions of *rs9926320* (**C**, **D**, *n* = 34,204) and *rs11126806* (**F**, **G**, *n* = 34,204) with cardiovascular conditions and an example of interaction between *rs7575796* and depression (**I**, **J**, *n* = 34,204) on cortical thickness are shown. The effect magnitude of these clinical conditions was modified by genotype

categories. A total of 49 significant interactions of genetic loci with cardiovascular conditions, 35 significant interactions with depression, and 11 significant interactions with both conditions (**E**, Supplementary Data 11–14, 21, 22) were found. Several polygenic risk scores (PRS) derived from the GWAS of regional thickness showed an interaction with cardiovascular health on executive function composite score in CLSA (**H**, Supplementary Data 15). In panels **D**, **G**, and **J**, means and 99.9% confidence intervals are plotted. All p-values are two-sided.

GxE interaction independently ($P_{FDR} < 0.1$). We then averaged the cortical thickness values within these regions and performed a second model on this aggregated metric, resulting in one composite model of cortical thickness for each genetic variant. Our GxE models identified 35 SNPs showing a significant interaction with depression, 49 SNPs showing a significant interaction with cardiovascular conditions, and 11 SNPs showing a significant interaction with both conditions on regional cortical thickness ($p < 1.3 \times 10^{-5}$, i.e., 0.05/220/18.1 effective comparisons, Fig. 4, Supplementary Fig. 3, Supplementary Data 11–14). Seven SNPs with depression interactions and 20 SNPs with cardiovascular conditions interactions were also significant at uncorrected $p < 0.05$ in a cross-validation analysis. Most interactions were qualitative in nature[39], with opposite effects of the SNP observed in cases compared to controls, although some quantitative interactions with the same effect direction but different effect magnitude in cases vs. controls were also found (Supplementary Fig. 4). Overall, these data indicate that some SNPs associated with cortical thickness also show interactions of a similar magnitude to their main effects. For example, compared to the base model including all covariates, the addition of the main effects of *rs11126806* explained 0.08% of the variance in regional thickness, while the addition of the interaction explained an extra 0.08% of the variance in regional thickness compared to the main effect model.

## Applying main effect genetic and G × E models to predict cognitive function in 25,000 mid- or late-life adults

Finally, building on our genetic and G × E interaction analyses of cortical thickness and WMH in the UKB imaging sample, we accessed data for an independent population-based cohort (CLSA) of over 25,000 adults between the ages of 45–86 with genetic, clinical, and neurocognitive assessment data. We first tested whether genetic variants associated with cortical thickness in UKB were also able to explain variation in cognitive performance in this sample. In our first set of analyses, 10 of the 215 genetic variants imputed with high quality in both samples had a significant effect on composite scores measuring memory or executive function, eight of which had a consistent direction with the effect on cortical thickness (Fig. 1C, F, $p_{FDR} < 0.1$). One of the genes was *COL4A2*, which has been linked to brain small vessel disease and vascular dementia[40–43]. Reassuringly, these cognition-associated variants in CLSA included two of three variants associated with fluid intelligence (*n* = 134,640) in UKB ($p_{FDR} < 0.1$, Supplementary Data 6, 7).

To examine the behavioral relevance of our findings and extend them to the CLSA sample, we calculated polygenic scores using summary statistics from our cortical thickness GWAS conducted in the UKB and tested whether they predicted better cognitive performance in CLSA. We focused on the GWAS-significant variants. Polygenic risk scores representing genetic determinants of thickness in the pars

orbitalis and pars triangularis of the inferior frontal gyrus predicted better executive function in those without hypertension but not in those with hypertension ($T = -2.75$, $p = 0.006$, $T = -2.80$, $p = 0.005$, $p_{FDR} < 0.1$) (Supplementary Data 15).

## Discussion

In genome-wide association analyses, we identified 367 significant associations with global and regional cortical thickness in over 34,500 mid- and late-life adults from the UKB in vivo. We probed the identified loci in more detail and found functionally significant genes whose transcription and allelic variation are associated with cortical thickness in postmortem data from an independent dataset. We showed that major depression and cardiovascular conditions moderate genetic effects on cortical thickness; specific genetic variants interacted with depression and cardiovascular conditions to exert additional risk for cortical thinning beyond either genes or environment alone. The relevance of our findings was further extended by the association of some of the same genetic variants with memory and executive function in over 25,000 mid- and late-life adults from CLSA. Finally, we showed the specific impact of GxE interactions, whereby a polygenic risk score representing higher cortical thickness in the inferior frontal gyrus predicted better executive function in participants without hypertension but not in those with hypertension.

The etiologies of Alzheimer's disease and related dementias (ADRD) are complex[44], with many studies individually attributing risk to genetic, epigenetic, or environmental factors[1,45,46]. For example, both depression and cardiovascular conditions are known risks for cognitive impairment[47] and dementia[1,2,48], and they can lead to cortical thinning, especially in the prefrontal cortex encompassing the insula. In turn, cortical atrophy or thinning is one of the best-established characteristics of cognitive decline and ADRD[49]. Thus, it is an important biological marker of brain health. While we found that many genetic variants affect brain and cognitive health independently from environmental factors, the effects of others are magnified in the presence of depression and cardiovascular conditions. Our findings of G × E interactions show that a substantial proportion of genome-wide significant variants (>12%) affect cortical thickness differently in the presence of two key modifiable risk factors for dementia: depression and cardiovascular conditions.

There are a number of modifiable risk factors for ADRD, including depression, cardiovascular conditions, smoking, physical inactivity, alcohol consumption, and hearing loss[1]. Mid-life and the start of late life are critical timepoints in the lifespan for addressing these risk factors[1]. However, it is unclear the extent to which treating depression or cardiovascular conditions could mitigate heritable risk for ADRD. A previous large-scale study has shown that a healthy lifestyle can decrease but not eliminate the genetic contribution of polygenic risk scores to dementia risk[50,51]. Moreover, G × E studies focusing on the APOE gene have also shown significant interactions with physical activity and alcohol consumption[44]. While G × E models are promising in disentangling the complex contributions of genetics and environment, more comprehensive and powerful G × E studies are needed. The UKB sample is ideal for assessing the G × E effects given its size and the age range (45–81) of its participants. In this population sample, there are very few people with dementia; however, it is possible to study biomarkers of preclinical AD by leveraging neuroimaging measures of brain health. In particular, cortical thinning is a good proxy measure of ADRD severity[15], including incipient dementia[52,53]. By building complex G × E models of cortical thickness as a measure of brain health, we pinpoint specific genetic variants interacting with depression or cardiovascular conditions. Our findings of interactive effects of polygenic scores for cortical thickness on cognitive function reinforce the conclusion that some genetic variants associated with cortical thickness differentially affect proxy measures of ADRD depending on the presence of cardiovascular conditions.

In view of previous large-scale analyses of the genetics of cortical thickness[24,25,54], including imaging phenotypes from the UKB[26], our

analyses offer several new findings. First, we analyzed regional cortical thickness without correcting for global thickness and explored in greater detail several novel loci. Most studies of cortical thickness in psychiatric disorders do not covary for global thickness[34], including ENIGMA meta-analyses[35,55] and large-scale UKB analyses[2,33], with some recent exceptions[56]. However, some studies covary for intracranial volume to control for head size, leading us to include this measure as a covariate. While our results are similar to those obtained in analogous analyses of the ENIGMA cohorts[24] (genetic correlation > 0.8), we found many distinct loci likely because our discovery sample included older adults without psychiatric conditions, unlike ENIGMA. We confirmed some of these associations in a smaller cohort of much older adults in ROS/MAP. Our heritability estimates were in line with previous studies of cortical thickness[24–26] and WMH[27]. We found genetic correlations between greater WMH volume and higher risk for ADHD, which itself was previously linked with higher levels of amyloid and tau[57], insomnia[58], and a lower intelligence quotient[59].

We show that some loci associated with cortical thickness are also associated with cognition and may play a functional role in cortical thickness via effects on gene expression. For example, rs11197843 was associated with higher anterior cingulate thickness and better memory performance and is an eQTL for gene SHTN1, which is important for axonogenesis[60]. Further, rs1562330, an eQTL for the gene STMN4, was associated with global and regional thickness and executive function. In turn, STMN4 expression also predicted cortical thickness, suggesting that this nervous system gene is central to the regulation of neural and cognitive processes. The role of STMN4 in microtubule polymerization has been suggested to increase neuronal complexity through evolution[36], which could be a potential mechanism through which this gene affects neuroimaging and cognitive phenotypes. It is also a critical gene in controlling axonal myelination in ADRD through the rearrangement of actin cytoskeleton[61]. In addition to STMN4, we have identified functionally significant genes linked to cortical thickness, including ARL17A, ARL17B, LRRC37A, and LRRC37A2, all of which were located on chromosome 17. Our findings in postmortem ROS/MAP data do not exclude the possibility that other genes, especially those identified in the eQTL analysis of our GWAS findings, play a functional role in regulating cortical thickness in younger populations or other brain regions.

Our study has several limitations. First, the UKB includes participants with a predominantly European ancestry. Larger well-phenotyped non-Caucasian samples are needed for our results to generalize across ancestries[62–64]. While PLINK-based linear modeling is computationally efficient, it may result in inflated type I error rates when population substructure is present in the study sample; future studies including more diverse populations would benefit from emerging multivariate[18] and mixed-effect models designed for biobank-scale analyses[65,66]. Second, current ROS/MAP data include relatively small numbers of participants with both neuroimaging and genetic data. While the sample size for the differential gene expression analysis is relatively modest, we were still able to identify several genes overlapping with the eQTL mapping of GWAS results. Third, we tested for G × E interactions with only two variables to reduce the number of false positives. A more comprehensive approach is needed to incorporate a range of environmental factors while tackling the challenge of multiple comparison corrections. Finally, cardiovascular conditions (Fig. 4) and depression[2] affect brain structure, and previous work has identified bi-directional associations between depression and white matter integrity using Mendelian Randomization[67]. Future studies should examine how neuroimaging phenotypes of brain structure, taken as a proxy measure for ADRD here[15,52], may also interact with other environmental factors to influence psychiatric conditions.

In conclusion, we identified novel genome-wide significant loci associated with cortical thickness, assessing their shared heritability, expression-based roles, and relevance to cognitive outcomes in midlife and older adults. We also found GxE interactions providing a template

for discovering more impactful variants affecting phenotypic variation in brain structure. Our findings suggest that treating depression and hypertension has the potential to mitigate some of the genetic risks for poor brain and cognitive outcomes. In addition, understanding an individual's underlying genetic architecture could clarify the risk for these poor outcomes and help to select preventative interventions.

## Methods

### Participants

Participants for primary analyses were part of the UKB cohort. We did not exclude participants with specific medical conditions. The overall number of participants with preprocessed MRI outputs was 40,669; however, we analyzed data from 35,846 White British[68] and European participants as defined by (https://pan.ukbb.broadinstitute.org/, Return 2442) given the low numbers of participants of other ancestries. Details on the UKB study protocols have been described[69,70]. The UKB study has obtained ethical approval from the National Health Service National Research Ethics Service (reference[11]:/NW/0382) and all participants provided informed consent. We compared our findings to the largest non-UKB univariate analysis of cortical thickness in the ENIGMA dataset[24].

For secondary analysis of postmortem brain tissue, we accessed data from the Religious Orders Study of the Rush Memory and Aging Project (ROS/MAP)[28]. These data were used to (1) replicate the associations between SNPs representing risk loci and cortical thickness, (2) replicate the interactive effects of risk loci with cardiovascular health issues on cortical thickness, and (3) examine direct transcriptome-wide associations between gene expression and regional cortical thickness. All ROS/MAP participants provided informed and repository consent and also signed the Anatomical Gift Act, and ethical approval was obtained from the institutional review board of Rush University Medical Center. A total of 202 ROS/MAP participants had both T1 MRI and genome-wide genotype data. Finally, we ran a transcriptome-wide differential expression analysis of MRI-derived cortical thickness in 66 ROS/MAP participants with both dorsolateral prefrontal cortex (DLPFC) RNA sequencing (RNAseq) and in vivo structural MRI data acquired prior to death.

Finally, we analyzed associations between genetic and cognitive function data in the CLSA, a sample of 25,387 older adults aged 45–86 from seven Canadian provinces[29–31]. Ethical approval was obtained from research ethics boards of all the participating institutions across Canada, and informed consent was obtained from all participants[71]. More information on participant characteristics can be found in Supplementary Data 1.

### Modifiable factors

Among the modifiable factors that determine risk for dementia, we focused on depression and cardiovascular health[1]. In the UKB, depression at the time of MRI scan was assessed using the Patient Health Questionnaire (PHQ-2)[2,72]; cardiovascular conditions were self-reported vascular/heart diagnoses from a physician (Data Field 6150, self-report questionnaire) and included heart attack ($n = 415$, 1.2%), angina ($n = 426$, 1.2%), stroke ($n = 227$, 0.7%) and hypertension ($n = 6514$, 18.2%), with most participants reporting none of these conditions ($n = 28,264$, 78.8%). Given the low frequency of heart attack, angina, and stroke, we combined all cardiovascular conditions into one cardiovascular variable encompassing at least one of these conditions.

### Data processing

*MRI*. Structural MRI data was processed using FreeSurfer by the UKB. We have used the cortical thickness outputs from the Desikan–Killiany parcellation[73] (UKB Data Fields 26755–26788 and 26856–26889), in the genome-wide association study (GWAS) analyses of bilateral thickness. In addition, we also analyzed the total volume of white matter hyperintensities (Data Field 25781). These imaging-derived phenotypes were generated by using pipelines developed and run on behalf of UKB

(https://biobank.ctsu.ox.ac.uk/crystal/crystal/docs/ brain_mri.pdf). For analyses of gene-by-environment (GxE) interactions based on variant prioritization, we used a more fine-grained Human Connectome Parcellation (HCP)[2,74]. Total intracranial volume was included as a covariate in all analyses. We excluded outliers with cortical thickness or WMH scores of ±4 s.d. from the mean in UKB analyses. We did not include the anterior temporal lobe since FreeSurfer reconstruction often fails in these regions in older adults. A similar approach was used to analyze MRI data from the ROS/MAP.

*Genetics*. Following previous work[27], we excluded related individuals with a Kinship-based INference for Gwas (KING) kinship coefficient ≥ 0.0884, keeping only one individual per group of up to second-degree relationships[75]. We also excluded individuals with mismatched genetically determined vs. self-reported sex. Quality control (QC) of genetic data was performed using PLINK version 2.00[76]. We filtered autosomal nonduplicate single-nucleotide variants with imputation information (INFO) score > 0.8, and with Hardy–Weinberg equilibrium $P > 10^{-10}$, missingness <5%, and minor allele frequency (MAF) > 0.1%. We focused on European UKB participants, with assignments and principal components (PCs) obtained from https://pan.ukbb.broadinstitute.org/. CLSA genetic data processing, including genotype QC and imputation to the TOPMed reference panel is described in more detail in previous studies[31]. CLSA QC procedures followed UK Biobank QC documentation[31].

### Main effect GWAS models

We used PLINK2 (https://www.cog-genomics.org/plink/2.0/assoc)[77] to independently test for additive allelic dosage associations with a) WMH, b) global cortical thickness calculated as the average thickness of the cortex, and c) 33 regional thickness in the large-scale UKB data. PLINK2 Models covaried for total intracranial volume, sex, age, study site, and the first 10 genomic PCs. Sensitivity analyses, including an expanded set of covariates, are shown in the Supplementary (Supplementary Data 20). Separate models were run on White British and non-White British European participants. We then used the METAL software package (z-scores method[78]) to perform meta-analyses across these models for variants with non-missing data in at least 10,000 individuals. Effect and reference allele, p-values, sample size, and direction of effect were included in the meta-analysis, with genomic control correction.

To replicate our findings, we tested for additive associations between 220 SNPs identified as genetic risk loci for cortical thickness in UKB analyses in ROS/MAP data. In each model, we covaried for total intracranial volume, sex, age, age[2], age × sex, ROS vs MAP site, and the first 10 genetic PCs (Matlab R2016a, *fitlme.m*). Since 55% of participants had repeated MRI measures, we included all data and fitted a random intercept for each participant (1| participant ID). Given the size of the ROS/MAP sample featuring MRI and SNP data ($n = 202$), we tested whether each of the UKB risk loci showed an association with the same (bilateral) regional thickness as in the UKB analyses (uncorrected $P < 0.05$). We also tested whether UKB risk loci showed a significant association with any of the 66 lateralized regions, setting the threshold to $p < 0.0015$ (0.05/33 regions for each SNP).

We examine the overlap between our genomic risk loci with those identified in the ENIGMA data[24] first by comparing the ENIGMA summary statistics for our loci and our summary statistics for the loci identified as significant in ENIGMA. Further, we used LDlink (https://ldlink.nih.gov/?tab=home, European reference population) to test whether any of the ENIGMA loci were in linkage disequilibrium (LD) with risk loci we identified in UKB (Supplementary Data 4).

### Locus definitions and SNP-to-gene mapping

We used the online FUMA (Functional Mapping and Annotation of Genome-Wide Association Studies, https://fuma.ctglab.nl/) toolkit with default settings for definitions of the lead SNPs and genomic risk loci[79,80]. Notably, we specified 1000 Genomes Phase 3 European as a

reference population for LD clumping, and the GWAS-level significance threshold was set to 5x10⁻⁸. For annotation of genomic risk loci in each GWAS, we used FUMA's SNP2GENE tool[81]. However, we used FUMA expression quantitative trait loci (eQTL) mapping to identify genes related to cortical thickness in our GWAS analyses for select phenotypes, namely for global cortical thickness and cortical thickness of caudal middle frontal and rostral middle frontal regions. These regions were selected since the ROS/MAP postmortem analysis was run on tissue from the dorsolateral prefrontal cortex (DLPFC), of which anatomical location approximately matches the caudal middle frontal and rostral middle frontal regions of the Desikan–Killiany atlas[73]. We examined eQTLs in tissue types of GTEx Brain cortex and GTEx Brain frontal cortex BA9 regions and set the false discovery rate threshold (FDR) at FDR < 0.05 to define significant eQTL associations.

### Heritability, genetic correlations, and gene ontology

We calculated heritability from GWAS summary statistics using the LD Score tool (LDSC; v 1.0.1 https://github.com/bulik/ldsc)[82]. Genetic correlations were estimated by comparing our summary statistics for global thickness and WMH (Supplementary Data 3 and 4) with summary statistics for schizophrenia, major depression, socioeconomic status, intelligence, insomnia, and Alzheimer's disease using LDSC[83] (Supplementary Data 18). We calculated the genetic correlation matrix for each pair of regional thickness summary statistics and used the Brain Connectivity Toolbox (BCT, https://sites.google.com/site/bctnet/) to assess network modularity based on this correlation matrix[84]. Finally, to identify biological processes implicated by GWAS analyses of regional cortical thickness, we used the *clusterProfiler* package in R (4.2.0) to conduct gene ontology (GO) overrepresentation analyses on position-mapped (ANNOVAR, Supplementary Data 10) genes, which maps several GO terms to several phenotypes in the same figure.

### Postmortem brain differential expression analysis

Postmortem bulk tissue RNAseq data for DLPFC from participants in ROS/MAP were accessed to identify genes whose expression was associated with cortical thickness. We analyzed a subset of brain tissue samples donated by participants who had also undergone antemortem MRI prior to death ($n = 66$). MRI acquisition details are provided in Supplementary Data 2. Full details on DLPFC RNAseq sample extraction, preprocessing, post-processing and QC, and statistical modeling have been previously published[85,86]. Paired-end sequencing was performed in three sets of a total of 13 batches, with an average depth of 50 million reads per sample. Reads were quantified according to the following pipeline: (1) fastq file QC was performed using FastQC v0.11.5 (default parameters), (2) STAR v2.5.3a was used to align reads (GRCh38.91 reference), (3) RSEM v1.2.31 was used to quantify expression from aligned BAM files, and (4) multiqc v1.5 was used to aggregate quality metrics from fastqc and Picard tools v2.17.4.

Differential expression analyses were performed on expected counts, aggregated across batches, using limma (v3.48.3) voom in R (v4.1.1). Model co-variates for inclusion in downstream analyses were determined by evaluating the effects of candidate variables on the top 20 principal components (PCs) of gene expression[86]. Brain cell type proportions, estimated by BRETIGEA[87], were also included as covariates, as described[86]. Robust linear modeling was used for differential expression, allowing up to 20,000 iterations to reach convergence. Significance of the effects for target outcomes in our multivariate models was performed using empirical Bayes moderation (eBayes function). *P*-values were adjusted using the FDR approach[88].

### Cell-type mapping with single-nucleus RNAseq

We used the Seattle Alzheimer's disease (SEA-AD) Brain Cell Atlas (https://portal.brain-map.org/explore/seattle-alzheimers-disease) to explore the cell types preferentially expressing genes identified in our GWAS and postmortem analyses of cortical thickness.

### Gene × Environment (G × E) interaction testing

We tested two approaches to variant prioritization that narrow the search for G × E interactions to a few select SNPs. First, we tested each of the 220 unique lead SNPs identified as genomic risk loci for global or regional cortical thickness. Second, we used the Deviation Regression Model to compute variance QTLs (vQTLs)[89]. However, no vQTLs were significant at the GWAS level of significance ($P < 5 × 10^{-8}$), thus leaving our initial set of variants of interest unchanged.

When testing for interactions, we chose the HCP parcellation[74], which includes 360 cortical regions and thus provides a more fine-grained map of the brain. For each SNP of interest, we computed an interaction map, controlling for site, age, sex, age × sex, age², total intracranial volume, and the first 10 principal components (*fitlm,m*, MATLAB R2016a). We applied $p < 0.01$ correction to each of the maps to identify regions with a significant SNP × depression or SNP × cardiovascular health interaction (Supplementary Fig. 3). We then averaged cortical thickness values across the significant regions ($p < 0.01$), resulting in one mean thickness variable for each SNP of interest that was entered into a model identical to the regional inter-action models. Given the substantial correlation among the 360 cortical thickness variables, we calculated a smaller number of "effective" comparisons following previous work[90–92] to guide multiple comparison correction. Specifically, we used principal component analysis to obtain the eigenvalues $\lambda_1,..., \lambda_p$ for the 360 cortical thickness phenotypes $p$, and calculated the number of effective phenotypes as $\frac{(\sum_{k=1}^{p} \lambda_k)^2}{\sum_{k=1}^{p} \lambda_k^2}$[91], obtaining 18.1 effective phenotypes. We further used cross-validation to test the robustness of the results from the whole sample in held-out data (Supplementary Section 2, Supplementary Data 21, 22).

### Gene effects on cognitive function

Using cognitive and genetic data from the CLSA, we aimed to test whether genetic risk loci for cortical thickness also affect cognitive function and whether polygenic scores for cortical thickness interact with hypertension to affect cognitive function. We first tested for associations of each variant from UKB GWAS analyses with composite scores of memory and executive function[29,30]. We also tested for associations of each variant from UKB GWAS with paired associates learning and fluid intelligence in UKB. Second, we calculated a polygenic score in CLSA for each brain region by multiplying the aligned effect allele dosage (0,1,2) with the summary statistics from the GWAS analysis (z-score) and adding them together to obtain the polygenic score. We then tested for main effects and interactions between the resulting polygenic score representing cortical thickness and the presence of hypertension on composite scores of memory and executive function. Models co-varied for age, sex, age², age × sex, years of education, and the first 10 genetic PCs.

### Reporting summary

Further information on research design is available in the Nature Portfolio Reporting Summary linked to this article.

## Data availability

The GWAS summary statistics generated in this study have been deposited with the GWAS catalog (https://www.ebi.ac.uk/gwas/) under GCST90399874–GCST90399911. Raw data are available under restricted access as follows. Data are available from the Canadian Longitudinal Study on Aging (www.clsa-elcv.ca) for researchers who meet the criteria for access to de-identified CLSA data. Data are available from the UK Biobank (application #61530) for researchers who meet the criteria for access to de-identified UK Biobank data. The UK Biobank is a uniquely powerful biomedical database. It aims to facilitate research in life sciences by providing multiscale data for a large number of participants. The UK Biobank legally binds the researchers

using the data not to publicly share UK Biobank data. Therefore, we are unable to share the data in a public repository. However, all data used here can be accessed by making a request with the UK Biobank. The UK Biobank has a dedicated portal for applying for data access here: https://www.ukbiobank.ac.uk/enable-your-research/apply-for-access. The use of UK Biobank data is not entirely free, but the data access costs are accessible to researchers. Researchers can submit a data request for ROS/MAP data to the Rush Alzheimer's Disease Center. SEA-AD data resource is publicly available at https://knowledge.brain-map.org/data as part of the Seattle Alzheimer's Disease Brain Cell Atlas Comparative Viewer. The summary statistics for significant genetic variants are also available in the Supplementary Information.

## Code availability

We share all code used in the manuscript on GitHub (https://github.com/peterzhukovsky/imaging_genetics; Zenodo https://doi.org/10.5281/zenodo.10895139 https://zenodo.org/records/10904367).

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

## Acknowledgements

R.O.S. and M.A.P. are supported by National Institute on Aging grants P30AG10161, P30AG72975, R01AG15819, and R01AG17917. U01AG46152, U01AG61356. R.O.S. and M.A.P. additional support from R01-AG059716. R.O.S. and M.A.P. resources can be requested at https://www.radc.rush.edu. Additional support from R01-AG059716. This research was made possible using the data/biospecimens collected by the Canadian Longitudinal Study on Aging (CLSA). Funding for the Canadian Longitudinal Study on Aging (CLSA) is provided by the Government of Canada through the Canadian Institutes of Health Research (CIHR) under grant reference: LSA 94473 and the Canada Foundation for Innovation, as well as the following provinces, Newfoundland, Nova Scotia, Quebec, Ontario, Manitoba, Alberta, and British Columbia. This research has been conducted using the CLSA dataset [Genome-wide Genetic Data Release version 3.0; Comprehensive Baseline Dataset Version 7.0], under Application Number [2006026]. The CLSA is led by Drs. Parminder Raina, Christina Wolfson and Susan Kirkland. The opinions expressed in this manuscript are the author's own and do not reflect the views of the Canadian Longitudinal Study on Aging.

## Author contributions

P.Z., A.N.V., and D.F. designed the study. P.Z., E.S.T. and D.F. analyzed the data and performed the research. P.Z., E.S.T., G.C., D.A.B., Y.W., T.J.H., D.A.P., B.H.M., A.N.V. and D.F. wrote the paper.

## Competing interests

Over the past 3 years, D.A.P. has received consulting fees from Albright Stonebridge Group, Boehringer Ingelheim, Compass Pathways, Engrail Therapeutics, Neumora Therapeutics (formerly BlackThorn Therapeutics), Neurocrine Biosciences, Neuroscience Software, Otsuka, Sunovion, and Takeda; he has received honoraria from the Psychonomic Society and American Psychological Association (for editorial work) and from Alkermes; he has received research funding from the Brain and Behavior Research Foundation, the Dana Foundation, Millennium Pharmaceuticals, Wellcome Leap MCPsych, and NIMH; he has received stock options from Compass Pathways, Engrail Therapeutics, Neumora Therapeutics, and Neuroscience Software. No funding from these entities was used to support the current work, and all views expressed are solely those of the authors. P.Z. was funded by the Canadian Institute of Health Research Postdoctoral Fellowship. G.C. was funded by the Alzheimer Society Canada Postdoctoral Fellowship. TH serves on the scientific advisory board for Vivid Genomics. D.F. is supported by the generous contributions from the Michael and Sonja Koerner Foundation and the Krembil Family Foundation. D.F. is also supported in part by the Centre for Addiction and Mental Health (CAMH) Discovery Fund and CIHR. B.H.M. holds and receives support from the Labatt Family Chair in Biology of Depression in Late-Life Adults at the University of Toronto. He currently receives or has received, within the past 5 years, research support from Brain Canada, the Canadian Institutes of Health Research, the CAMH Foundation, the Patient-Centered Outcomes Research Institute (PCORI), the US National Institute of Health (NIH), Capital Solution Design LLC (software used in a study funded by the CAMH Foundation), and HAPPYneuron (software used in a study funded by Brain Canada). Within the past 5 years, B.H.M. has also received research support from Eli Lilly (medications for an NIH-funded clinical trial) and Pfizer (medications for an NIH-funded clinical trial). He has been an unpaid consultant to Myriad Neuroscience. A.N.V. currently receives funding from CIHR, the NIH, the National Sciences and Engineering Research Council (NSERC), the CAMH Foundation, and the University of Toronto. E.S.T. was funded by the Ontario Graduate Scholarship. The remaining authors declare no competing interests.
