## [Peer Review File · Nature Communications]

Genetic influences on brain and cognitive health and their interactions with cardiovascular conditions and depressionEditorial Note: Parts of this Peer Review File have been redacted as indicated to remove third-party material where no permission to publish could be obtained.

REVIEWER COMMENTS

Reviewer #1 (Remarks to the Author):

This is a comprehensive study of association of cortical thickness (regional and global), white matter hyperintensities (WMH) and genetics, in multiple cohorts (UK Biobank, ROSMAP, CLSA). 35 genetic loci were found to be associated with global thickness and WMH, many of them de novo. A few hundreds were associated with regional thickness measures. Significant correlations between WMH and insomnia, ADHD, intelligence were found. Variable heritability of thickness across brain regions is reported. Network modularity analysis identified 3 distinct modules. Gene enrichment analyses identified biological processes related to cell growth and development. GeneXenvironment interactions were investigated for depression and cardiovascular factors and 19 associations were found. Genetic variants associated with cortical thickness in UKB were found to be associated with 8 cognitive measures in CLSA. In general, it is a strong study adding to our knowledge of associations between certain brain structural features and genetics as well as gene expression in an elderly population. I am therefore supportive of publication. My enthusiasm is somehow tempered by the relatively weak genetic associations found, and by fact that the clinical variables they chose are not necessarily modifiable (e.g. depression, insomnia), as would be factors like smoking, obesity, exercise etc. More generally, I have a bit of trouble seeing how the findings can inform patient management, clinical trials, and interventions.

Some specific points are discussed below:

1. The problem itself is very interesting and clinically meaningful. Identifying modifiable risk factors is crucial for overall brain health. However, the two factors studied here are less likely modifiable. Depression and cardiovascular themselves are complicated conditions. The major conclusion here is that these two conditions mediate the genetic effects on cortical

thickness and WMH. However, there is abundant evidence that imaging-derived endophenotypes (e.g., cortical thickness) cause brain diseases, such as depression (reference paper here: <https://www.nature.com/articles/s41467-020-16022-0>). In addition, the endophenotype model is well-defined in Psychiatry:

[https://ajp.psychiatryonline.org/doi/10.1176/appi.ajp.160.4.636?url_ver=Z39.88-](https://ajp.psychiatryonline.org/doi/10.1176/appi.ajp.160.4.636?url_ver=Z39.88-2003&rfr_id=ori:rid:crossref.org&rfr_dat=cr_pub)

2003&rfr_id=ori:rid:crossref.org&rfr_dat=cr_pub

0pubmed Therefore, studying complex modifiable risk factors such as depression, we cannot ignore the bi-directional effects (associations or causality). The current study needs more evidence to support their claim in the title -- the word "influence" strongly suggests the direction of the effect.

2. Also, the definition of depression and cardiovascular condition in UKBB is self-reported questionnaire, making the imaging population very heterogeneous to study such complex interactions. The authors should clearly state what other conditions (based on ICD10) they excluded or did not exclude for population selections.

3. The first paragraph's statement in the Abstract is not precise/correct: "Approximately 40% of dementia cases result from modifiable risk factors related to lifestyle and environment." No reference was given, but I believe this is from Livingston et al., 2020 published in Lancet (<https://www.ncbi.nlm.nih.gov/pmc/articles/PMC7392084/>). I read the paper, and their original statement is: 'Modifying 12 risk factors might prevent or delay up to 40% of dementias'. The words "might prevent or delay" does not mean 40% of dementia cases RESULT FROM these modifiable factors.

4. Covariates correction in the GWAS model: the covariates considered are not sufficient. The very first few papers of GWAS on IDP from UKBB have carefully investigated this. Including the ones from groups like Smith et.al. in Oxford and Bingxin Zhao et.al. at UNC/Penn. For instance, the interaction between age x sex, age-squared, and age-squared x sex, the first 40 genetic PCs, the scan position, etc. The reviewer did not rerun all experiments, but the authors should at least take these previous papers as a reference.

5. The genetic QC step "Variants with MAF <0.01, low imputation quality ($R^2 < 0.8$), or were available in less than 10,000 individuals were excluded." (--mind) should be done together with other QC steps. Why did the author separately do this? For "low imputation quality ($R^2 < 0.8$)" and "imputation information (INFO) score > 0.8", what are the differences? The description of the genetic QC should be stated more clearly.

6. The replication of the GWAS findings is not clear. Did the author run the GWAS in all

baseline scans, and then replicate the 220 SNPs? The statement is weak, and the P-value threshold is too liberal: "Given the size of the ROS/MAP sample featuring MRI and SNP data (n=202), we tested whether each UKB SNP showed a trend association with the same (bilateral) regional thickness as in the UKB analyses (P<0.1)"

7. The authors interchangeably use 220 SNPs (line 148) and 220 unique genetic loci (in the abstract). Please refer to the original FUMA paper to be sure of the definition of a genetic/genomic locus, and be precise.

8. These two sentences are not clear: "We used the online FUMA (Functional Mapping and Annotation of Genome-Wide Association Studies, <https://fuma.ctglab.nl/>) with default settings for definitions of the lead SNPs and genomic risk loci" vs. "For the annotation of the genomic risk loci in each of the GWAS, we used the ANNOVAR software package". FUMA automatically defined the genomic loci using underlying methods. If the authors run post-GWAS analysis on FUMA online platform - which I believe so - then authors should not state, "we used the ANNOVAR software package."

9. The study used the Science GWAS paper from ENIGMA as a reference for comparisons, and the replication rate is very low: "Among the 22 loci significantly associated with global thickness, only four were also identified as genome-wide significant by the most recent ENIGMA GWAS 24, likely since we did not include global thickness as a covariate and due to the more homogeneous makeup of our UKB sample compared to ENIGMA." Of note, the Science paper itself did include the UK Biobank participants. In addition, they found "***indicating consistent genetic architecture between the 49 ENIGMA cohorts and data collected from a single scanner at the primary UK Biobank imaging site."

10. Again, the authors interchangeably used loci vs. SNPs when reporting the omnibus GWAS (22 independent loci) vs. regional GWAS (345 associations). It is ok to do so, but it should be precisely specified.

11. The heritability estimates obtained in this study is much lower than the one observed in previous studies

1. <https://www.nature.com/articles/s41586-018-0571-7/figures/1>

2. <https://www.nature.com/articles/s41588-019-0516-6>

The authors should explain this.

13. The PRS calculation is not clear. The authors stated, "To examine the behavioral

relevance of our findings and extend them to the CLSA sample, we calculated polygenic scores using summary statistics from our cortical thickness GWAS conducted in the UKB, and tested whether they predicted better cognitive performance in CLSA. We focused on the GWAS-significant variants." Choose the top signals may cause the "winner's curse" and failed to generalize to unseen data.

14. How was intracranial volume determined? Freesurfer and other packages are limited, especially in aging populations, in which atrophy can be confounded by ICV change. Show data elucidating ICV accuracy, including longitudinal measures from the 55% of participants with longitudinal data

15. Some analyses appear to need cross-validation. For example, in the "Gene x Environment (GxE) interaction testing" section, regions/parcels with significant GXE interaction were determined. The cortical thickness was then averaged selectively into the significant regions/parcels, and the resultant measures were input into another interaction model. This step seems questionable to me, as it can grossly inflate the estimated interactions of the second step, unless tested on a separate, independent cohort (i.e. the lumped cortical thickness associated with each SNP was maximally fitted to this particular dataset first, and then tested on the same dataset)

16. Although GWAS findings identified SNPs linked to 25 genes influencing cortical thickness, only 4 of them were found to have functional role, based on post-mortem gene expression analyses. I am puzzled by this inference, as many of these genes might have been active during brain development and wouldn't be expected to be found in the ROSMAP post-mortem analysis of an elderly population.

17. I might have missed it, but the cognitive associations of the UKB GWAS findings in CLSA were not tested in UKB itself---are these genes also associated with similar cognitive measures in UKB, when available?

Reviewer #2 (Remarks to the Author):

The authors set out to improve upon existing studies of the genetics of imaging-based brain phenotypes as relates to dementia risk. They conduct GWAS in the UK Biobank, finding new loci impacting these traits and exploring regional specificity of the genetic signals within the brain. They augment these results with transcriptional data from ROS/MAP and extend

them to cognitive function phenotypes in CLSA. Finally, they perform GxE tests, finding variant-specific interactions with CVD and depression impacting the same phenotypes (in UKB and ROS/MAP) and polygenic score interactions impacting cognitive phenotypes (in CLSA).

This study is a great example of “not just a GWAS”. The authors test genetic associations with disease-relevant phenotypes and use a combination of functional annotation and follow-up analyses to understand the relevant genes and the relevant brain regions involved. This type of mechanistic detail is helpful in addressing the variant-to-function challenge that faces the genetics community. I don’t personally have specific experience with brain or cognitive phenotypes, but it appears that this would be a valuable contribution to the genetic literature in this space. My most substantial comments relate to the changing use of significance thresholds throughout the manuscript and the centrality of GxE to the primary scientific story. Some other improvements could also be helpful for ease of reading, such as more complete referencing of each Figure panel and each Supplementary Table in the text itself.

Major points:

- The use of p-value and FDR thresholds changes across the manuscript, and can feel a bit arbitrary at times. Ideally, these thresholds would be more homogeneous across the manuscript (for example, $5e-8$ for all genetic analyses, $FDR < 0.05$ for all RNA-seq or multi-region analyses, and $p < 0.05$ for all replication analyses). If the authors feel strongly that all of the specific, differing thresholds are appropriate, then it might at least be helpful to have a holistic discussion somewhere summarizing why these differing choices are appropriate for the various analyses. A few specific questions about these thresholds:
 - It’s a bit difficult to interpret the proportion of UKB discovery GWAS hits “replicated” at $p < 0.1$ in ROS/MAP by itself. Can the authors provide a statistical test (simple proportion test of observed vs. expected replications) to back up the significance of this replication? How many replicate and what are the results of this same significance test if a standard nominal threshold is used ($p < 0.05$)?
 - The multi-threshold approach for Bonferroni correction in the GxE analysis (SNP #-only correction versus region-and-SNP # correction) is straightforward and transparent. One additional possibility is to correct for an “effective” number of regions based on the inter-

region correlation, which would generate a single, region-and-SNP Bonferroni threshold between the current two thresholds. See Wang et al. 2019 Sci. Adv. or Westerman et al. 2022 Nat. Comms. for an example of this “effective number of phenotypes” strategy.

- The importance of GxE in this overall scientific story should be clarified. The title, abstract, and intro seem to motivate the study based on the GxE concept, but a fair amount of the results involve uncovering and characterizing genetic main effects.

- The flow of the Intro might be able to be tweaked so it is clear that the motivation is both (1) increasing our understanding of the genetic architecture of CT and WMH, and (2) understanding the presence of GxE. One straightforward suggestion might be to just take the final sentence of paragraph 2 (about interactions) and move it to the end of paragraph 3 (so the narrative is: biology -> genetics -> GxE).

- Can the authors draw a more direct link between the SNP-specific interactions impacting brain structure outcomes (in UKB and ROS/MAP) and the PRS interactions impacting cognitive performance (in CLSA)? Connections between the biology uncovered or the ways that these analyses reinforce each other would be helpful in giving the reader something more concrete to take away.

- It might be valuable to visualize one or more of these interactions with depression/CVD as the primary variable. For example, for the plots where genetic effects have different sign across disease categories, it might be that the disease effect on brain structure is directionally-consistent with only magnitude changes across genotype categories. This is especially relevant given that the Discussion emphasizes how these diseases act as modifiable risk factors for ADRD.

Specific comments (in order of the manuscript):

- A bit more clarification could be helpful in describing the UKB subsets used (European from Pan-UKBB based on genetics, White British vs. non-White British based on a questionnaire, etc.) and why related choices were made (e.g., defending the choice to run separate models in *ethnic* subgroups of genetically-determined European ancestry).

- Given that these UKB genetic data were imputed, was the variant missingness filter (>10k individuals available) relevant?

- Methods-GxE: GxE models adjust for the first 10 principal components – of what? Genetic PCs, or cortical region thickness PCs, or something else?

- Results-GWAS: By “an association with the exact same locus”, do the authors mean “exact same variant or a variant in moderate-strong LD...”?
- Results-heritability: When referencing the heritability of entorhinal and anterior prefrontal regions, should it read “ $h=0.1$ ” rather than “ $r=0.1$ ”?
- Results-heritability: How do these cross-region genetic correlations (and their modules) compare to a similar analysis using phenotypic correlations? Are there any insights uniquely contributed by the genetics for this particular question? One option would be to use one half of the existing heatmap to show phenotypic correlations.
- Results-gene expression: “...we performed computational fine mapping of genetic loci from our GWAS using eQTL data...”. It would be helpful to have more detail here. Was a direct fine mapping done (in or outside of FUMA) and did it actually require eQTL data (i.e., functionally-informed fine mapping)? Are the authors referring to colocalization analysis instead? Or, is this intended to indicate rather a direct lookup of GWAS SNPs for significance in eQTL datasets? Even if some of this is described/encoded in the FUMA pipeline, it would be helpful to provide more details in this manuscript since this component of the pipeline is important.
- Results-Chr17: I’m hesitant about the language of “...only LRR37A, LRR37A2. and ARL17A, ARL17B play a functional role...”, given the limited sample size and biological (post-mortem samples) limitations of these RNA-seq results. Also note the period instead of comma in the sentence above.
- Results-Chr17: Is the “anatomical anterior-posterior ordering” observation based on an analysis and/or expected based on prior literature? Or was this an ad hoc visual observations made by the authors?
- Results (general): Conditional analyses, controlling for the most-significant variant(s), could be helpful in reinforcing some of the statements about multiple signals in a locus (beyond simply having LD below the fairly liberal threshold of $r^2 < 0.4$).
- Results-GxE: “combing” -> “combining”
- Fig. 4: Panels E,F may not be necessary – without genomic context or other annotations, this visual information doesn’t seem to add much compared to a simple table.
- Results-GxE: How were the attributable proportions of variance explained by G & GxE (for example, for rs11126806) calculated?
- Results-GxE: Am I correctly understanding that the ROS/MAP replication of SNP-specific

GxEs was achieving p-values <0.0021 even with only 66 samples? Where did that p-value threshold come from?

- Results-GxE: It would be helpful to have clarification or citations related to the use of the terms “multiplicative” and “additive”. It seems like the authors are using these terms to describe what are often referred to as “qualitative” (changing effect signs) and “quantitative” (changing only magnitudes) interactions. The terms “multiplicative” and “additive” are usually reserved for distinct types of statistical tests, often in the context of binary outcomes.

Dec 6, 2023

We would like to thank the Reviewers for taking the time to provide thorough and insightful comments. Below we respond to each of the Reviewer's comments (Reviewer feedback is presented in *italics*, while our responses are shown in **blue**. Quoted text from the manuscript or supplementary is shown in black, with added text underlined). We think that the manuscript is improved in terms of the robustness of analyses, amount of detail regarding the methods, and interpretation of the effects we found based on the Reviewer's comments.

Reviewer #1 (Remarks to the Author):

This is a comprehensive study of association of cortical thickness (regional and global), white matter hyperintensities (WMH) and genetics, in multiple cohorts (UK Biobank, ROSMAP, CLSA). 35 genetic loci were found to be associated with global thickness and WMH, many of them de novo. A few hundreds were associated with regional thickness measures. Significant correlations between WMH and insomnia, ADHD, intelligence were found. Variable heritability of thickness across brain regions is reported. Network modularity analysis identified 3 distinct modules. Gene enrichment analyses identified biological processes related to cell growth and development. GeneXenvironment interactions were investigated for depression and cardiovascular factors and 19 associations were found. Genetic variants associated with cortical thickness in UKB were found to be associated with 8 cognitive measures in CLSA. In general, it is a strong study adding to our knowledge of associations between certain brain structural features and genetics as well as gene expression in an elderly population. I am therefore supportive of publication. My enthusiasm is somehow tempered by the relatively weak genetic associations found, and by fact that the clinical variables they chose are not necessarily modifiable (e.g. depression, insomnia), as would be factors like smoking, obesity, exercise etc. More generally, I have a bit of trouble seeing how the findings can inform patient management, clinical trials, and interventions.

We thank the Reviewer for their helpful feedback and positive evaluation. We believe that the manuscript is greatly improved following the Reviewer's feedback. Specifically, we have added several new analyses which appear both in the main text and Supplementary Information: 1) First, we quantitatively assess the impact of including different covariates in our GWAS models in accordance with the state-of-the-art in modeling imaging-derived phenotypes, 2) Second, we cross-validate our GxE analyses to quantify generalizability and overfitting, and 3) Third, we improve comparative analyses of our identified genomic risk loci and those identified in published ENIGMA data by Grasby et al. We have also improved the Title, Introduction, and Discussion based on provided feedback to address the bi-directional links between imaging and depression, the potential clinical utility of our findings, and other issues related to clarity and context.

Some specific points are discussed below:

1. The problem itself is very interesting and clinically meaningful. Identifying modifiable risk factors is crucial for overall brain health. However, the two factors studied here are less likely modifiable. Depression and cardiovascular themselves are complicated conditions. The major conclusion here is that these two conditions mediate the genetic effects on cortical thickness and WMH. However, there is abundant evidence that imaging-derived endophenotypes (e.g., cortical thickness) cause brain diseases, such as depression (reference paper here: <https://www.nature.com/articles/s41467-020-16022-0>). In addition, the endophenotype model is well-defined in Psychiatry: https://ajp.psychiatryonline.org/doi/10.1176/appi.ajp.160.4.636?url_ver=Z39.88-2003&rfr_id=ori:rid:crossref.org&rfr_dat=cr_pub 0pubmed Therefore, studying complex modifiable risk factors such as depression, we cannot ignore the bi-directional effects (associations or causality). The current study needs more evidence to support their claim in the title -- the word "influence" strongly suggests the direction of the effect.

We thank the Reviewer for bringing this important point to our attention and have added the following section to address the bi-directional associations in the limitations section of the Discussion (newly added text is underlined):

“A more comprehensive approach is needed to incorporate a range of environmental factors while tackling the challenge of multiple comparison correction. Finally, cardiovascular conditions (Figure 4) and depression¹ affect brain structure and previous work has identified bi-directional associations between depression and white matter integrity using Mendelian Randomisation². Future studies should examine how neuroimaging phenotypes of brain structure, taken as a proxy measure for ADRD here, may also interact with other environmental factors to influence psychiatric conditions.”

We have also updated the title (previously “Modifiable risk factors for dementia interact with genetic risk to influence brain and cognitive health”) following the Reviewer’s feedback:

Title: “Genetic influences on brain and cognitive health and their interactions with modifiable risk factors for dementia”

2. Also, the definition of depression and cardiovascular condition in UKBB is self-reported questionnaire, making the imaging population very heterogeneous to study such complex interactions. The authors should clearly state what other conditions (based on ICD10) they excluded or did not exclude for population selections.

We thank the Reviewer for highlighting the need to clarify the inclusion criteria and we now clarify this at the start of the Methods.

“Participants for primary analyses were part of the UKB cohort. We did not exclude participants with specific medical conditions.”

We use the self-report definitions of the conditions to ensure a sufficient number of participants with depression and cardiovascular conditions, as having smaller case groups and genetic variants with relatively low prevalence of the effect allele would make interaction testing very challenging.

3. The first paragraph's statement in the Abstract is not precise/correct: "Approximately 40% of dementia cases result from modifiable risk factors related to lifestyle and environment." No reference was given, but I believe this is from Livingston et al., 2020 published in Lancet (<https://www.ncbi.nlm.nih.gov/pmc/articles/PMC7392084/>). I read the paper, and their original statement is: 'Modifying 12 risk factors might prevent or delay up to 40% of dementias'. The words "might prevent or delay" does not mean 40% of dementia cases RESULT FROM these modifiable factors.

While we could not include the reference in the abstract due to journal guidelines, we do cite it at the start of the Introduction. However, the Reviewer is correct, and we have updated the abstract statement to reflect the conclusions of those findings more accurately, as below:

“Approximately 40% of dementia cases could be prevented or delayed~~result from~~ by modifiable risk factors related to lifestyle and environment.”

4. Covariates correction in the GWAS model: the covariates considered are not sufficient. The very first few papers of GWAS on IDP from UKBB have carefully investigated this. Including the ones from groups like Smith et.al. in Oxford and Bingxin Zhao et.al. at UNC/Penn. For instance, the interaction between age x sex, age-squared, and age-squared x sex, the first 40 genetic PCs, the scan position, etc. The reviewer did not rerun all experiments, but the authors should at least take these previous papers as a reference.

We thank the Reviewer for this suggestion and have performed a series of new analyses to address the important issue of covariate selection. First, we performed new GWAS analyses for A) global cortical thickness and B) three of 33 individual regions of interest (those specifically most important for interaction testing shown in Figure 4H) following suggested guidelines from Smith et al., Zhao et al. and Alfaro-

Almagro et al. In these four new GWAS, we found that adding an age x sex interaction term and using 40 genomic PCs instead of 10 had a relatively small impact on our findings; co-varying for the extended set of interaction and non-linear terms reduced the number of significant loci identified at the genome-wide level, suggesting somewhat reduced power. However, concordance of observed effects overall was extremely high. Given previous GWAS studies (e.g. Hofer et al. 2020, *Nature Communications*) have used the more limited set of covariates, and given that we aimed to maximize power to identify candidate SNPs for interaction testing, we include the 10 PCs, age, sex, and site as covariates in the main GWAS analysis. Importantly, we co-vary for age, sex, age² and age x sex interactions alongside 10 PCs in all interaction testing, both at the SNP level and at the PRS level. Full details of these additional analyses are provided below and also in the revised manuscript in Supplementary Section 1, Supplementary Figures 7 and 8, and Supplementary Table 20.

“Supplementary Information

1. GWAS re-analysis including a more expansive set of covariates

To test the impact of including different sets of covariates, we re-ran a selection of four GWAS including age x sex, age², as covariates in addition to age, sex, site and total intracranial volume as included in the main analysis and 40 genetic principal components instead of the 10 PCs included in the primary analysis^{26,90,91}. The GWAS outcomes we selected were global cortical thickness and thickness of inferior frontal gyrus pars orbitalis, inferior frontal gyrus pars triangularis, and the supramarginal gyrus, given the prominence of these regions identified by our GxE interaction testing. We assessed 1) the impact of the expanded covariate set on risk loci mapped by FUMA and 2) the impact of the expanded covariate set on the wider distribution of effect sizes (Z-statistics from the meta-analysis of GWAS) and p-values for each cortical thickness phenotype.

In Supplementary Table 20 we show the summary statistics for the genomic risk loci in the main analyses and in the sensitivity analyses alongside each other. Among the 49 genomic risk loci from the four GWAS, we found that 7 did not replicate in the expanded covariate analysis, whereby replication required us to find a SNP from the genomic risk locus with LD $r^2 > 0.6$ with the SNP from the genomic risk locus from the main analyses. Further, we found three new variants using the expanded set of covariates.

Second, we show the impact of expanding the set of covariates on Z-statistics and p-values for these four GWAS in the Supplementary Figures 7 and 8, respectively. The correlations between the Z-statistics (including Z-statistics with uncorrected $p < 0.05$) from the main analysis and from the analyses including an expanded set of covariates was very high ($r > 0.999$, Supplementary Figure 7). Further, Bland-Altman plots⁹⁰ showed that the inclusion of an expanded set of covariates had an appreciable, but comparatively small impact on the significance of findings in the four GWAS (Supplementary Figure 8).”

“References

26. Smith, S. M. et al. An expanded set of genome-wide association studies of brain imaging phenotypes in UK Biobank. *Nat. Neurosci.* 24, 737–745 (2021).

90. Alfaro-Almagro, F. et al. Confound modelling in UK Biobank brain imaging. *Neuroimage* 224, 117002 (2021).

91. Zhao, B. et al. Genome-wide association analysis of 19,629 individuals identifies variants influencing regional brain volumes and refines their genetic co-architecture with cognitive and mental health traits. *Nat. Genet.* 51, 1637–1644 (2019).”

“Supplementary Figure 7. Correlations between Z-statistics (derived from METAL) for all SNPs showing an association at $p < 0.05$ with cortical thickness in main analyses (y axis) and in the supplementary analyses including an extended set of covariates for each of the four GWAS.

Supplementary Figure 8. Bland-Altman (BA) plots showing how associations between SNPs and cortical thickness in the four selected GWAS are affected by the addition of an extended covariate set. If the addition of extra covariates does not strongly affect the associations, the points should follow a horizontal distribution around $y=0$. On the other hand, if there are substantial differences in the significance of SNP-cortical thickness associations between the standard and extended set of covariates, we would see a shift away from the $y=0$ line. A strong downward shift would suggest reduced significance and potentially loss of genomic risk loci, while a strong upward shift would suggest the emergence of new loci that were not significant with the standard set of covariates. The plots show that the change in $-\log_{10}$ of P-values was relatively small, with the highest shift of -1.5 for global thickness and supramarginal thickness. A shift of 1 would correspond to a 10-fold change in p-value, for instance from 0.01 to 0.001.

5. The genetic QC step "Variants with $MAF < 0.01$, low imputation quality ($R^2 < 0.8$), or were available in less than 10,000 individuals were excluded." (--mind) should be done together with other QC steps. Why did the author separately do this? For "low imputation quality ($R^2 < 0.8$)" and "imputation information ($INFO$) score > 0.8 ", what are the differences? The description of the genetic QC should be stated more clearly.

We thank the Reviewer for spotting this issue – we have mistakenly duplicated elements of our QC procedure in the "Genetics" and in the "Main effect GWAS models" sections of the methods. We have deleted the duplicate in the "Main effect GWAS models" as below, and have clarified that the 10,000 participant filter was applied at the METAL meta-analysis stage:

“PLINK2 Models covaried for total intracranial volume, sex, age, study site, and the first 10 genomic PCs. Separate models were run in White British and non-White British European participants. Variants with MAF ≤ 0.01 , low imputation quality ($R^2 < 0.8$), or were available in less than 10,000 individuals were excluded. We then used the METAL software package (z-scores method⁶) to perform meta-analyses across these models for variants with non-missing data in at least 10,000 individuals.”

6. *The replication of the GWAS findings is not clear. Did the author run the GWAS in all baseline scans, and then replicate the 220 SNPs? The statement is weak, and the P-value threshold is too liberal: "Given the size of the ROS/MAP sample featuring MRI and SNP data (n=202), we tested whether each UKB SNP showed an association with the same (bilateral) regional thickness as in the UKB analyses (P<0.1)"*

We have clarified the ROS/MAP replication analysis in the Methods section – the Reviewer’s description of the procedure is correct. We now also apply a more consistent threshold across the different replication and validation analyses (P<0.05). As a consequence, we saw a somewhat lower replication rate in ROS/MAP compared to P<0.1.

“Methods

Given the size of the ROS/MAP sample featuring MRI and SNP data (n=202), we tested whether each of the of the UKB SNP risk loci showed an association with the same (bilateral) regional thickness as in the UKB analyses (uncorrected P<0.05)”

“Results

Further, 38 21 associations from the discovery analysis also passed a liberal replication threshold (p<0.05 P<0.1) in a cis-replication analysis of the ROS/MAP cohort.”

7. *The authors interchangeably use 220 SNPs (line 148) and 220 unique genetic loci (in the abstract). Please refer to the original FUMA paper to be sure of the definition of a genetic/genomic locus, and be precise.*

We thank the Reviewer for pointing out this important distinction and have updated the relevant references to genomic risk loci throughout the manuscript. We therefore replace references to SNPs when those SNPs were identified as a top SNP representing an independent genomic risk locus as mapped by FUMA. For example, in the GxE section, we now write:

“We tested two approaches to variant prioritization that narrows the search for GxE interactions to a few select SNPs. First, we tested each of the 220 unique lead SNPs identified as genomic risk loci for global or regional cortical thickness.”

To clarify, our cortical thickness GWAS of global thickness and 33 regional thickness phenotypes have returned 367 genomic risk loci across these phenotypes. We carried the top SNP for each of these 367 risk loci as mapped by FUMA forward to secondary analyses of cognitive effects and GxE interactions. Since we obtained the 367 risk loci from 34 individual GWAS of intercorrelated cortical thickness phenotypes, some of the loci were overlapping (e.g. rs11126806 was the top SNP for the genomic risk locus associated with global thickness and also for a risk locus associated with inferior parietal thickness). We therefore used 220 unique SNPs across these 367 loci. We also provide clumped results across all GWAS in Supplementary Table 5 (and clumped results for the interaction SNPs in Supplementary Tables 12 and 13) but chose to test all unique SNPs for interactions to maximize the candidate SNP pool.

8. *These two sentences are not clear: "We used the online FUMA (Functional Mapping and Annotation of Genome-Wide Association Studies, <https://fuma.ctglab.nl/>) with default settings for definitions of the lead SNPs and genomic risk loci" vs. "For the annotation of the genomic risk loci in each of the GWAS, we used the ANNOVAR software package". FUMA automatically defined the genomic loci using underlying methods.*

If the authors run post-GWAS analysis on FUMA online platform - which I believe so - then authors should not state, "we used the ANNOVAR software package."

We thank the Reviewer for identifying this framing issue – we indeed used FUMA’s default methods for mapping SNPs to functional annotations and amended the Methods to reflect this.

“For annotation of genomic risk loci in each GWAS, we used FUMA’s Gene2Func tool package⁴⁴.”

We also add a supplementary Section to expand further on the eQTL analyses:

“Supplementary Section 6. eQTL analyses in FUMA

We used FUMA's eQTL mapping package with default settings to map SNPs identified in select GWAS to genes⁷. FUMA maps SNPs to genes based on a significant eQTL association in GTEx brain cortex tissue and GTEx Frontal Cortex BA9 tissue data. We selected these tissue types given that the phenotypes of interest were cortical regions, and we included BA9 specifically given its overlap with the ROS/MAP post-mortem tissue. Significant SNP-gene pairs were defined using false discovery rate-corrected $p < 0.05$. The eQTL analysis thus allows us to identify genes, whose expression was significantly associated with allelic variation at the SNP in GTEx v8.”

*9. The study used the Science GWAS paper from ENIGMA as a reference for comparisons, and the replication rate is very low: "Among the 22 loci significantly associated with global thickness, only four were also identified as genome-wide significant by the most recent ENIGMA GWAS 24, likely since we did not include global thickness as a covariate and due to the more homogeneous makeup of our UKB sample compared to ENIGMA." Of note, the Science paper itself did include the UK Biobank participants. In addition, they found "***indicating consistent genetic architecture between the 49 ENIGMA cohorts and data collected from a single scanner at the primary UK Biobank imaging site."*

We thank the Reviewer for bringing up the comparative analysis between our UKB findings and the ENIGMA GWAS paper by Grasby et al. To clarify, Grasby et al. used only 5,096 participants from the UKB, whereas we were able to include over 34,000 UKB participants. However, we have improved our comparative analysis by testing each of the genomic risk loci we identified for LD with all the genomic risk loci that Grasby et al. report using the LDlink package from the NIH (<https://ldlink.nih.gov/>). We report the highest LD SNPs and the LD r^2 for our loci and Grasby’s loci in Supplementary Table 1. We also report the number of loci that showed a high LD $r^2 > 0.4$.

“We examine the overlap between our genomic risk loci with those identified in the ENIGMA data⁸ first by comparing the ENIGMA summary statistics for our loci and our summary statistics for the loci identified as significant in ENIGMA. Further, we used LDlink (<https://ldlink.nih.gov/?tab=home>, European reference population) to test whether any of the ENIGMA loci were in linkage disequilibrium (LD) with risk loci we identified in UKB (Supplementary Table 4).”

~~<https://pubs.broadinstitute.org/mammals/haploreg/haploreg.php>. We listed all SNPs that were in LD with our loci ($r^2 > 0.4$, 1000G Phase 1 European reference population) and cross-checked the ENIGMA loci with the resulting list”~~

10. Again, the authors interchangeably used loci vs. SNPs when reporting the omnibus GWAS (22 independent loci) vs. regional GWAS (345 associations). It is ok to do so, but it should be precisely specified.

Following the Reviewer’s feedback, we have updated the references to SNPs vs. genomic risk loci in this section of the results. We report the genomic risk loci from FUMA for all our GWAS’s; we used “associations” previously because the genomic risk loci for different cortical regions may overlap. In reporting these results we follow Grasby et al. who also reported genomic risk loci for each region alongside each other even if the genomic risk locus was the same SNP. We also provide clumped results across all GWAS’s in Supplementary Table 5.

“In 33 regional GWAS analyses, we identified an additional 345 risk loci for cortical thickness of 33 cortical regions (at uncorrected $p < 5.0 \times 10^{-8}$) and provide results clumped across all cortical thickness GWAS in Supplementary Table 5.”

11. The heritability estimates obtained in this study is much lower than the one observed in previous studies

1. <https://www.nature.com/articles/s41586-018-0571-7/figures/1>

2. <https://www.nature.com/articles/s41588-019-0516-6>

The authors should explain this.

We found that the heritability estimates in Grasby et al. ranged from 0.08 to 0.26 (Supplementary Table 7, uncorrected for global thickness). In our analyses we found heritability to range from between 0.10 to 0.29 (Supplementary Table 16), which is slightly higher than Grasby's findings but more in line with the higher heritability values in the UKB figure in reference 1 of the Reviewer's comment, whereby cortical thickness heritability values range between 0.05 and 0.4. We attribute the slightly higher estimates to a more homogeneous sample in UKB which comprises mostly healthy older adults, unlike the component patient groups with psychiatric conditions included in ENIGMA.

13. The PRS calculation is not clear. The authors stated, "To examine the behavioral relevance of our findings and extend them to the CLSA sample, we calculated polygenic scores using summary statistics from our cortical thickness GWAS conducted in the UKB, and tested whether they predicted better cognitive performance in CLSA. We focused on the GWAS-significant variants." Choosing the top signals may cause the "winner's curse" and failed to generalize to unseen data.

We selected the top GWAS hits (i.e. genomic risk variants from FUMA) to ensure the PRSs are relevant to GxE interaction analyses which prioritized the same top GWAS hits, and the association analyses with cognitive function. While we considered including more variants at less stringent significance thresholds, this avenue would make us test genetic signal that is different from the variants we focus on in the main GWAS and GxE analyses. Importantly, we selected the GWAS hits for PRS calculation from UKB and tested them in unseen data in CLSA. That is, all our PRS testing is out-of-sample since we use UKB to select the genetic variants and weights for the PRS and test them in CLSA.

14. How was intracranial volume determined? Freesurfer and other packages are limited, especially in aging populations, in which atrophy can be confounded by ICV change. Show data elucidating ICV accuracy, including longitudinal measures from the 55% of participants with longitudinal data

Indeed, we used FreeSurfer-derived ICV measures. While FreeSurfer-derived reconstruction of cortical surface sometimes fails (especially in anterior temporal regions), it is generally very robust and has been included in ABCD pipelines (Hagler et al 2019, <https://www.ncbi.nlm.nih.gov/pmc/articles/PMC6981278/>) and in fmriprep (Esteban et al 2018, <https://www.nature.com/articles/s41592-018-0235-4>) as well as in UKB neuroimaging pipelines. To account for major reconstruction failure, we excluded extreme outlier participants (more than 4 standard deviations from the mean, similar to the approach used by Anderson et al. (2020, PNAS). Nonetheless, we agree that demonstrating the reliability of ICV over time is helpful, and now we show baseline vs. follow-up total ICV from FreeSurfer in Supplementary Figure 11, as suggested:

“Supplementary Figure 11. Total intracranial volume (TIV) at baseline vs follow-up shows an extremely high degree of consistency ($r=0.986$, $n=4,408$ with baseline and follow-up data).

15. Some analyses appear to need cross-validation. For example, in the “Gene x Environment (GxE) interaction testing” section, regions/parcels with significant GXE interaction were determined. The cortical thickness was then averaged selectively into the significant regions/parcels, and the resultant measures were input into another interaction model. This step seems questionable to me, as it can grossly inflate the estimated interactions of the second step, unless tested on a separate, independent cohort (i.e. the lumped cortical thickness associated with each SNP was maximally fitted to this particular dataset first, and then tested on the same dataset)

We thank the Reviewer for this helpful suggestion. We have added a cross-validation analysis to test the robustness of the interactions and present it in the main text and supplementary information. We now also include more brain regions when testing for interactions as can be seen in Supplementary Figure 3 and Figure 4 to ensure we retain enough power in split samples in cross-validation analyses.

“Supplementary Section 2. Cross-validation analysis of GxE interactions

In order to test the robustness of our GxE findings, we repeated all analyses using cross-validation. For each of the 220 candidate SNPs, we first split our sample in 10 folds. For each split, we fit linear models in 90% of the sample, creating a GxE interaction brain map (featuring 360 regions) for depression and cardiovascular disease. We then thresholded each brain map at uncorrected $p < 0.01$ in the inner fold (90% of the sample), and proceeded to test for interactions in the outer fold, i.e. the held-out 10% of the data. For regions that passed the $p < 0.01$ threshold in the inner fold, we averaged the cortical thickness values in the outer fold and obtained an interaction p-value. Where no significant regions were identified, we used global cortical thickness as an outcome. We repeated the process 10 times, testing each combination of held-out outer fold and inner fold data. This cross-validation procedure allowed us to test the robustness of the regions with an interaction in held-out data. As a result, for each SNP, we obtained a p-value for each of the cross-validation splits. We combined each set of 10 p-values across the splits using Fisher’s method⁹. An overview of the approach is shown in Supplementary Figure 3.

The summary statistics for whole-sample analyses and cross-validation analyses for cardiovascular disease and depression are shown in Supplementary Tables 21 and 22, respectively. We found that many of the SNPs that passed the stringent significance threshold in the main analyses ($p < 0.05/220/18.1$ effective comparisons) also passed the uncorrected $p < 0.05$ threshold in the cross-validation analyses. We found that 20 of the 60 SNPs with a significant interaction with cardiovascular disease in the main analyses were also significant at uncorrected $p < 0.05$ following cross-validation. Among 46 SNPs with a significant interaction with depression in the main analyses, 7 SNPs were also significant at uncorrected $p < 0.05$ following cross-validation.”

We have also updated Supplementary Figure 3 to provide a schematic overview of the cross-validation approach:

“Supplementary Figure 3. Gene-by-environment interaction testing workflow and results.

We also updated Figure 4 accordingly in response to feedback by Reviewers 1 and 2 and reproduce it below.

16. Although GWAS findings identified SNPs linked to 25 genes influencing cortical thickness, only 4 of them were found to have functional role, based on post-mortem gene expression analyses. I am puzzled by this inference, as many of these genes might have been active during brain development and wouldn't be expected to be found in the ROS/MAP post-mortem analysis of an elderly population.

We thank the Reviewer for pointing this out and agree that this limiting factor deserves more attention. First, we have removed the text from our Results section suggesting that these are the “only” genes with likely functional roles, which we agree is not a sound interpretation given the nature of the ROS/MAP data:

“eQTL mapping using expression data from the frontal cortex and BA9 linked these SNPs to 25 genes, ~~but~~ and our postmortem RNAseq findings ~~suggest that only show that~~ LRRC37A, LRRC37A2, and ARL17A, ARL17B play a functional role, as their expression was associated with caudal middle frontal thickness of the PFC in the independent ROS/MAP cohort.”

Second, we now emphasize the fact that the eQTL mapping performed by FUMA provides a particular “flavour” of indirect evidence of association between genes and cortical structure when compared to the direct differential gene expression (DGE) analyses we performed with measured cortical thickness in ROS/MAP. Therefore, not only does the small ROS/MAP sample size limit our power for detecting DLPFC-expressed genes correlated with cortical structure in older adults *de novo*, but any lack of overlap with FUMA/eQTL-mapped genes does not necessarily indicate a replication problem. The average (SD) age of the 129 donors with frontal cortex BA9 RNAseq data in the GTEx v7 sample as downloaded by the authors was 58.0 (10.5) years old, while the average age of the ROS/MAP sample included in our analyses was 75.9 (7.0). Given that gene expression is influenced by age, we agree with the Reviewers that other genes beyond those identified in the DGE in ROS/MAP may also be regulating cortical thickness.

To clarify these limitations and extent of reasonable inference, we have amended the Discussion in two places:

“Second, current ROS/MAP data include relatively small numbers of participants with both neuroimaging and genetic data and offers low power for the replication analysis of the GWAS findings. While the sample size for the differential gene expression analysis is relatively modest, we were still able to identify several genes overlapping with the eQTL mapping of GWAS results.”

“In addition to STMN4, we have identified functionally significant genes linked to cortical thickness, including ARL17A, ARL17B, LRRC37A, LRRC37A2, all of which were located on chromosome 17. Our findings in post-mortem ROS/MAP data do not exclude the possibility that other genes, especially those identified in the eQTL analysis of our GWAS findings, play a functional role in regulating cortical thickness in younger populations or other brain regions.”

17. I might have missed it, but the cognitive associations of the UKB GWAS findings in CLSA were not tested in UKB itself--are these genes also associated with similar cognitive measures in UKB, when available?

We thank the Reviewer for this suggestion. We have tested the 220 risk loci from the UKB for associations with cognition both in CLSA and in UKB. In the original manuscript we show the associations of the unique 220 risk loci from cortical thickness GWAS's and the loci from the white matter hyperintensity GWAS with executive function (CLSA), memory (CLSA), paired associates learning (UKB) and fluid intelligence (UKB). In the original submission, we show the complete results in Supplementary Table 6 alongside FDR corrected P-values, with several associations passing $p_{FDR} < 0.1$ threshold. We also had reported these findings in the main text:

“Results

Finally, building on our genetic and GxE interaction analyses of cortical thickness and WMH in the UKB imaging sample, we accessed data for an independent population-based cohort (CLSA) of over 25,000 adults between the ages of 45-86 with genetic, clinical, and neurocognitive assessment data. We first tested whether genetic variants associated with cortical thickness in UKB were also able to explain variation in cognitive performance in this sample. In our first set of analyses, 10 of the 215 genetic variants imputed with high quality in both samples had a significant effect on composite scores measuring memory or executive function, eight of which had a consistent direction with the effect on cortical thickness (Figure 1C, 1F, $p_{FDR} < 0.1$). One of the genes was *COL4A2*, which has been linked to brain small vessel disease and vascular dementia¹⁰⁻¹³. Reassuringly, these cognition-associated variants in CLSA included two of three variants associated with fluid intelligence (n=134,640) in UKB ($p_{FDR} < 0.1$, Supplementary Table 6).”

We also updated Figure 1 to show the SNPs associated with cognitive function in both CLSA and UKB (panels C, F):

Reviewer #2 (Remarks to the Author):

The authors set out to improve upon existing studies of the genetics of imaging-based brain phenotypes as relates to dementia risk. They conduct GWAS in the UK Biobank, finding new loci impacting these traits and exploring regional specificity of the genetic signals within the brain. They augment these results with transcriptional data from ROS/MAP and extend them to cognitive function phenotypes in CLSA. Finally, they perform GxE tests, finding variant-specific interactions with CVD and depression impacting the same phenotypes (in UKB and ROS/MAP) and polygenic score interactions impacting cognitive phenotypes (in CLSA).

This study is a great example of “not just a GWAS”. The authors test genetic associations with disease-relevant phenotypes and use a combination of functional annotation and follow-up analyses to understand the relevant genes and the relevant brain regions involved. This type of mechanistic detail is helpful in addressing the variant-to-function challenge that faces the genetics community. I don’t personally have specific experience with brain or cognitive phenotypes, but it appears that this would be a valuable contribution to the genetic literature in this space. My most substantial comments relate to the changing use of significance thresholds throughout the manuscript and the centrality of GxE to the primary scientific story. Some other improvements could also be helpful for ease of reading, such as more complete referencing of each Figure panel and each Supplementary Table in the text itself.

We thank the Reviewer for their positive evaluation and helpful feedback. We have updated the Introduction, Title and Discussion to better reflect the focus of analyses on both GWAS and GWAS follow-up analyses and also on GxE interaction testing. We have improved the interaction analysis in two important ways: first, we have changed the significance threshold to the number of effective comparisons following the Reviewer’s suggestion, and second, we have added a cross validation analysis following feedback from Reviewer #1. We have also amended Figure 4, improving the visualization of specific interactions, following the Reviewer’s comments and added several Supplementary sections. We now provide an overview of the significance thresholds and have amended them to be more uniform, i.e. 5×10^{-8} for GWAS, $P_{FDR} < 0.1$ for most non-GWAS analyses, and uncorrected $p < 0.05$ for validation/replication analyses. We provide more information on each of these changes in response to the Reviewer’s points below.

Major points:

1. The use of p-value and FDR thresholds changes across the manuscript, and can feel a bit arbitrary at times. Ideally, these thresholds would be more homogeneous across the manuscript (for example, 5×10^{-8} for all genetic analyses, $FDR < 0.05$ for all RNA-seq or multi-region analyses, and $p < 0.05$ for all replication analyses). If the authors feel strongly that all of the specific, differing thresholds are appropriate, then it might at least be helpful to have a holistic discussion somewhere summarizing why these differing choices are appropriate for the various analyses. A few specific questions about these thresholds:

We thank the Reviewer for this opportunity to improve the clarity of our approach to multiple testing correction in the manuscript. To summarize, we adopt significance thresholds of 5×10^{-8} for GWAS and $p_{FDR} < 0.5$ for the RNAseq analyses of cortical thickness in ROS/MAP as suggested by the Reviewer. We used $p_{FDR} < 0.1$ for analyses of cognitive function in CLSA/UKB for the following two reasons: First, previous studies with comparable sample sizes to the one we analyzed report smaller magnitude of genetic effects on cognitive function (eg only 3 loci found for verbal–numerical reasoning by Davies et al. (2016, Mol Psych) in over 34,000 participants). Therefore, to ensure sufficient power to detect effects we have used the 0.1 threshold. Second, our CLSA analyses were exploratory and aimed to follow up the UKB discovery analyses. We acknowledge these point in the section below:

“Supplementary Section 3. Summary of multiple testing correction approach

We conducted several sets of analyses with different significance thresholds chosen for the purpose of the analysis. In our GWAS studies, we used the conventional discovery threshold of 5×10^{-8} . In Supplementary Table 1 we also indicate which variants pass the more stringent $5 \times 10^{-8}/34 = 1.5 \times 10^{-9}$ threshold, which accounts for 34 individual region cortical thickness GWAS using the Bonferroni method.

Next, in extension of our UKB GWAS findings, we analyzed the much smaller ROS/MAP sample. In the first set of analyses of ROS/MAP genotype data (n=202), we tested for associations between the 220 SNPs identified as genomic risk loci for cortical thickness in UKB analyses and cortical thickness in ROS/MAP. When testing whether loci associated with cortical thickness of a specific region in our GWAS were associated with the thickness of the same region in ROS/MAP, we adopted a liberal pseudo-replication threshold of $p < 0.05$. When testing whether genomic risk loci for a specific cortical thickness trait from the UKB were associated with any cortical thickness in any region in ROS/MAP, we adopted a more stringent threshold of 0.05/33, again using the Bonferroni method, since the space of tests was now expanded 33 times for each GWAS-implicated gene. In the second set of analyses of the ROS/MAP RNASeq data (n=66), we have tested for differential gene expression associated with cortical thickness of the rostral middle frontal and caudal middle frontal regions. We aimed to identify genes that were associated with cortical thickness both in the ROS/MAP differential gene expression analyses at a liberal pseudo-replication threshold of $p < 0.05$ and in the UKB GWAS and eQTL analyses.

Next, we tested for the associations between cognitive function, measured in both CLSA and UKB, and the 220 unique SNPs representing genomic risk loci identified in our GWAS. We applied false discovery rate (FDR) correction $p_{FDR} < 0.1$ to each cognitive outcome. Each cognitive outcome (memory in CLSA, executive function in CLSA, paired associated learning in UKB, and fluid intelligence in UKB) was treated as a separate analysis, with multiple comparison correction applied within each analysis. We also generated 34 polygenic risk scores in CLSA using the summary statistics from our UKB GWAS, as described in the Methods. We tested for associations between memory and executive function in CLSA and these polygenic scores, reporting significant results at $p_{FDR} < 0.1$ and suggestive effects at uncorrected $p < 0.05$. We used a threshold of $p_{FDR} < 0.1$ in the analyses of cognitive function for two reasons. First, our secondary analyses of cognitive function aimed to further explore and follow up on the GWAS findings in UKB; therefore, we aimed to have the most power to detect genetic effects on cognition. Second, previous studies of cognitive function of comparable sample size in UKB have identified only few independent SNP signals after LD clumping for verbal-numeric reasoning⁹³ and much larger samples (n>300,000) were needed to identify a large number of SNPs linked to general cognitive function⁹⁴. Since our CLSA sample included just 25,387 participants, we acknowledge this as a limitation and have applied the threshold of $p_{FDR} < 0.1$. CLSA participants are very healthy older adults, with very few participants with cognitive impairment^{14,15}, reducing the variability in cognitive performance.

Finally, we conducted a series of targeted GxE analyses. In our main GxE analyses in UKB, we corrected for the number of SNPs tested (220) and the number of effective cortical thickness comparisons (18.1), resulting in a conservative threshold of $p < 1.3 \times 10^{-5}$ ($p < 0.05/220/18.1$). We then conducted a cross-validation analysis whereby the cortical thickness regions that we included in the interaction testing were generated in a separate split of the UKB data. We report the cross-validation effects at uncorrected $p < 0.05$. When generating the mean regional cortical thickness value from the interaction brain maps (Supplementary Figure 3), we used a threshold of $p < 0.01$ to ensure a substantial number of regions are included in testing the interactions for each of the 220 candidate SNPs. In the GxE analyses of polygenic risk scores, we tested for GxE interactions between cortical thickness polygenic scores for the 33 regions and executive function and memory in CLSA, reporting significant results at $p_{FDR} < 0.1$ and suggestive effects at uncorrected $p < 0.05$ following the rationale above. A limitation of our results is that our interaction findings are significant at $p_{FDR} < 0.1$ but not at $p_{FDR} < 0.05$.”

2. It's a bit difficult to interpret the proportion of UKB discovery GWAS hits “replicated” at $p < 0.1$ in ROS/MAP by itself. Can the authors provide a statistical test (simple proportion test of observed vs. expected replications) to back up the significance of this replication? How many replicate and what are the results of this same significance test if a standard nominal threshold is used ($p < 0.05$)?

Following the Reviewer's feedback, we have changed the threshold to 0.05 finding that 21 SNPs were replicated and have included the 95% confidence intervals for proportions in the Supplementary:

“For this region-to-region replication, we adopted a liberal threshold of $p < 0.05$. Among 317 variants overlapping between UKB and ROS/MAP, 21 variants were significant at $p < 0.05$, corresponding to 6.6%, with the 95% confidence intervals indicating that this proportion was significantly higher than zero [3.2%, 8.8%].”

3. The multi-threshold approach for Bonferroni correction in the GxE analysis (SNP #-only correction versus region-and-SNP # correction) is straightforward and transparent. One additional possibility is to correct for an “effective” number of regions based on the inter-region correlation, which would generate a single, region-and-SNP Bonferroni threshold between the current two thresholds. See Wang et al. 2019 Sci. Adv. or Westerman et al. 2022 Nat. Comms. for an example of this “effective number of phenotypes” strategy.

We thank the Reviewer for this suggestion and have implemented it in the main text and Figure 4 of the manuscript:

“Methods

...

Given the substantial correlation among the 360 cortical thickness variables, we calculated a smaller number of “effective” comparisons following previous work^{16–18} to guide multiple comparison correction. Specifically, we used principal component analysis to obtain the eigenvalues $\lambda_1, \dots, \lambda_p$ for the 360 cortical thickness phenotypes p , and calculated the number of effective phenotypes as $\frac{(\sum_{k=1}^p \lambda_k)^2}{\sum_{k=1}^p \lambda_k^2}$, obtaining 18.1 effective phenotypes. We report the results at P thresholds of 2×10^{-4} (0.05/220 SNPs) and 6×10^{-7} (0.05/220/360 SNPs and regions).”

Supplementary Figure 3:

We also included more brain regions in the summary score as can be seen in Supplementary Figure 3 and Figure 4 to ensure we retained enough power in the split samples in cross-validation analyses.

“Figure 4. Gene-by-environment interactions on cortical thickness. (A, B) Cardiovascular conditions reduced cortical thickness (CT) in the insular regions and increased the total volume of white matter hyperintensities (WMH). Examples of interactions of *rs9926320* (C,D) and *rs11126806* (F,G) with cardiovascular conditions and an example of interaction between *rs7575796* and depression (I,J) on cortical thickness are shown. The effect magnitude of these clinical conditions was modified by genotype categories. A total of 49 significant interactions of genetic loci with cardiovascular conditions and 35 significant interactions with depression and 11 significant interactions with both conditions (E, Supplementary Tables 11-14, 21, 22) were found. Several polygenic risk scores (PRS) derived from the GWAS of regional thickness showed an interaction with cardiovascular health on executive function composite score in CLSA (H, Supplementary Table 15). In panels D, G, and J means and 99.9% confidence intervals are plotted. All p-values are two-sided.”

4. The importance of GxE in this overall scientific story should be clarified. The title, abstract, and intro seem to motivate the study based on the GxE concept, but a fair amount of the results involve uncovering and characterizing genetic main effects. The flow of the Intro might be able to be tweaked so it is clear that the motivation is both (1) increasing our understanding of the genetic architecture of CT and WMH, and (2) understanding the presence of GxE. One straightforward suggestion might be to just take the final sentence of paragraph 2 (about interactions) and move it to the end of paragraph 3 (so the narrative is: biology -> genetics -> GxE).

We thank the Reviewer for this helpful suggestion and have adjusted the flow of the introduction by moving the mention of interactions to the third paragraph.

We have also updated the title (previously “Modifiable risk factors for dementia interact with genetic risk to influence brain and cognitive health”) following the Reviewer’s feedback:

Title: “Genetic influences on brain and cognitive health and their interactions with modifiable risk factors for dementia”

6. *Can the authors draw a more direct link between the SNP-specific interactions impacting brain structure outcomes (in UKB and ROS/MAP) and the PRS interactions impacting cognitive performance (in CLSA)? Connections between the biology uncovered or the ways that these analyses reinforce each other would be helpful in giving the reader something more concrete to take away.*

Following the Reviewer’s suggestion, we have added the following section to the Discussion (to and Supplementary:

“Discussion

...By building complex GxE models of cortical thickness as a measure of brain health, we pinpoint specific genetic variants ~~of which impact on markers of brain or cognitive health is modulated by the presence of~~ interacting with depression or cardiovascular conditions. Our findings of interactive effects of polygenic scores for cortical thickness on cognitive function reinforce the conclusion that some genetic variants associated with cortical thickness differentially affect proxy measures of ADRD depending on the presence of cardiovascular conditions.”

“Supplementary Section 8. Notable GxE SNPs of interest

We found some SNPs that showed a significant interaction with cardiovascular conditions on cortical thickness in the UKB to also contribute to the polygenic scores that showed an interaction with cardiovascular conditions on executive function in CLSA. In particular, rs77690628 and rs3200031 have been associated with cortical thickness of the pars triangularis and pars orbitalis in our analyses and in previous studies 92 and also showed a significant interaction with cardiovascular conditions on cortical thickness. These SNPs have been mapped to the PP2R2A gene, which plays a role in negative control of cell growth and division (<https://www.genecards.org/cgi-bin/carddisp.pl?gene=PPP2R2A>).”

7. *It might be valuable to visualize one or more of these interactions with depression/CVD as the primary variable. For example, for the plots where genetic effects have different sign across disease categories, it might be that the disease effect on brain structure is directionally consistent with only magnitude changes across genotype categories. This is especially relevant given that the Discussion emphasizes how these diseases act as modifiable risk factors for ADRD.*

We thank the Reviewer for this clarification and have updated Figure 4 to show the effects of disease category by genotype (please find the Figure 4 reproduced in response to Reviewer #1 above). We have moved the interaction plots showing the effects of genotype by disease category from Figure 4 to Supplementary Figure 9.

“Supplementary Figure 9. Example interactions for rs11126806 (A), rs9926320 (B), and rs7575796 (C), showing the effect of genotype category stratified by the presence of cardiovascular conditions or depression.

Specific comments (in order of the manuscript):

- A bit more clarification could be helpful in describing the UKB subsets used (European from Pan-UKBB based on genetics, White British vs. non-White British based on a questionnaire, etc.) and why related choices were made (e.g., defending the choice to run separate models in *ethnic* subgroups of genetically-determined European ancestry).

We thank the Reviewer for this suggestion and have added a supplementary section to expand on these choices. Briefly, running separate GWAS and meta-analyzing the results is a strength of our analyses since it allows for separate effect estimation in different ancestries and thus allows for more model flexibility. It was also applied in Grasby et al. (2020, Science) with different ancestries.

“Supplementary Section 5. UKB sample description

In the UKB, we have conducted GWAS on participants of European ancestry, defined by genetic data from the Pan-UKBB consortium (<https://pan.ukbb.broadinstitute.org/downloads/index.html>). Extending our findings to non-European populations such as those included in the large, representative AllOfUs study (<https://allofus.nih.gov/>) will be critical in the future. We split the European sample into White British and Non-White British participants based on previous studies of the UKB data¹⁹. We used the definitions provided by Tanigawa et al¹⁹, who state the following criteria: first, participants self-reported as having white British ancestry; second, their data was used to compute genetic principal components; third, they were not marked as outliers for heterozygosity and missing rates; fourth they did not show putative sex chromosome aneuploidy and finally, they had at most 10 putative third-degree relatives. While criteria 2-5 were related to genetic QC, the defining variable was self-reported ancestry. Running separate GWAS for the different European ancestries and meta-analyzing the results is a strength of our analyses, since it allows for separate effect estimation in different ancestries and thus for more model flexibility.”

- Given that these UKB genetic data were imputed, was the variant missingness filter (>10k individuals available) relevant?

We apologize for the confusion regarding this threshold, which was also noticed by Reviewer #1. This missingness filter is part of the METAL meta-analysis pipeline, which ensures that there aren't variants with exceptionally high missingness across subjects. While the genotype data were imputed, not all variants can be imputed with equal quality across samples, and METAL thus introduces the requirement of non-missingness per-variant. For our meta-analyses, the filter ensures that across all participants there are at least 10,000 participants with available allele data. It was not relevant for our analyses since we had over 27,000 white British participants and over 7,000 non white British European participants with genotyping data. We have now clarified this in the methods:

“PLINK2 Models covaried for total intracranial volume, sex, age, study site, and the first 10 genomic PCs Separate models were run in White British and non-White British European participants. ~~Variants with MAF <0.01, low imputation quality ($R^2 < 0.8$), or were available in less than 10,000 individuals were excluded.~~ We then used the METAL software package (z-scores method⁶) to perform meta-analyses across these models for variants with non-missing data in at least 10,000 individuals.”

• *Methods-GxE: GxE models adjust for the first 10 principal components – of what? Genetic PCs, or cortical region thickness PCs, or something else?*

We thank the Reviewer for pointing this out and now consistently refer to the genetic PCs throughout the manuscript:

“In each model we covaried for total intracranial volume, sex, age, age², age x sex, ROS vs MAP site, and the first 10 genetic PCs”

• *Results-GWAS: By “an association with the exact same locus”, do the authors mean “exact same variant or a variant in moderate-strong LD...”?*

We thank the Reviewer for bringing this framing issue to our attention and have updated the manuscript following this suggestion:

“either as an association with the exact same variant locus or a variant locus in moderate-strong LD ($r^2 > 0.4$).”

• *Results-heritability: When referencing the heritability of entorhinal and anterior prefrontal regions, should it read “h=0.1” rather than “r=0.1”?*

We thank the Reviewer for pointing out this error and have corrected it in the Results:

“showed lowest heritability ($h^2_{SNP}=0.1$, Supplementary Table 16).”

• *Results-heritability: How do these cross-region genetic correlations (and their modules) compare to a similar analysis using phenotypic correlations? Are there any insights uniquely contributed by the genetics for this particular question? One option would be to use one half of the existing heatmap to show phenotypic correlations.*

We thank the Reviewer for this suggestion and have updated Figure 2 to include both genetic and phenotypic correlations in the lower half of the heatmap as shown below:

Figure 2. Heritability and genetic correlations. (A) Heritability estimated using UKB GWAS results ($n > 34,500$) was lowest for the entorhinal, frontal pole, anterior cingulate, and orbitofrontal cortices. (B) Genetic correlations between cortical regions without covarying for global cortical thickness are shown above the diagonal, while phenotypic correlations between regional cortical thickness pairs are shown below

the diagonal. (C) Genetic correlations between global thickness and white matter hyperintensity with Attention Deficit and Hyperactivity Disorder (ADHD), schizophrenia (SCZ), major depressive disorder (MDD), Sensitivity to Environmental Stress and Adversity (SESA), intelligence (INTEL), insomnia (INSOM) and Alzheimer's disease (AD). *two-sided $p < 0.007$."

• *Results-gene expression*: "...we performed computational fine mapping of genetic loci from our GWAS using eQTL data...". It would be helpful to have more detail here. Was a direct fine mapping done (in or outside of FUMA) and did it actually require eQTL data (i.e., functionally-informed fine mapping)? Are the authors referring to colocalization analysis instead? Or, is this intended to indicate rather a direct lookup of GWAS SNPs for significance in eQTL datasets? Even if some of this is described/encoded in the FUMA pipeline, it would be helpful to provide more details in this manuscript since this component of the pipeline is important.

Following the Reviewer's suggestion, we have updated the main text and supplementary to provide more information on the eQTL mapping

"In parallel, we performed computational fine mapping of genetic loci from our GWAS using eQTL data from GTEx (using frontal cortex and BA9 references in FUMA, Supplementary Section 6)."

"Supplementary Section 6. eQTL analyses in FUMA"

We used FUMA's eQTL mapping package with default settings to map SNPs identified in the GWAS to genes42. FUMA maps SNPs to genes based on a significant eQTL association from GTEx brain cortex tissue and GTEx Frontal Cortex BA9 tissue data. We selected these tissue types given that the phenotypes of interest were cortical regions, and we included BA9 specifically given its overlap with the ROS/MAP post-mortem tissue. Significant SNP-gene pairs were defined using false discovery rate-corrected $p < 0.05$. The eQTL analysis thus allows us to identify genes, whose expression was significantly associated with allelic variation at the SNP in GTEx v8.

• *Results-Chr17*: I'm hesitant about the language of "...only *LRRC37A*, *LRRC37A2*. and *ARL17A*, *ARL17B* play a functional role...", given the limited sample size and biological (post-mortem samples) limitations of these RNA-seq results. Also note the period instead of comma in the sentence above.

This is an important point which was also raised by Reviewer #1. We have amended the sentence as follows in response to both Reviewers' comments in the Results:

"eQTL mapping using expression data from the frontal cortex and BA9 linked these SNPs to 25 genes, and our postmortem RNAseq findings ~~suggest that only~~ show that *LRRC37A*, *LRRC37A2*, and *ARL17A*, *ARL17B* play a functional role, as their expression was associated with caudal middle frontal thickness of the PFC in the independent ROS/MAP cohort."

We have also added the following clarification based on the Reviewer's suggestions in the limitations section of the Discussion:

"Second, current ROS/MAP data include relatively small numbers of participants with both neuroimaging and genetic data and offers low power for the replication analysis of the GWAS findings. While the sample size for the differential gene expression analysis is relatively modest, we were able to identify several genes overlapping with the eQTL mapping of the GWAS results."

Finally, we have explicitly mentioned that the eQTL analysis may have identified genes that also play a functional role in the Discussion (added text underlined):

"In addition to *STMN4*, we have identified functionally significant genes linked to cortical thickness, including *ARL17A*, *ARL17B*, *LRRC37A*, *LRRC37A2*, all of which were located on chromosome 17. Our findings in post-mortem ROS/MAP data do not exclude the possibility that other genes, especially those

identified in the eQTL analysis of our GWAS findings, play a functional role in regulating cortical thickness in younger populations or other brain regions.”

The average (SD) age of the 129 donors with frontal cortex BA9 RNASeq data in the GTEx v7 sample as downloaded by the authors was 58.0 (10.5) years old, while the average age of the ROS/MAP sample included in our analyses was 75.9 (7.0). Given that gene expression is influenced by age, we agree with the Reviewers that other genes beyond those identified in the DGE in ROS/MAP may be also be regulating cortical thickness.

• *Results-Chr17: Is the “anatomical anterior-posterior ordering” observation based on an analysis and/or expected based on prior literature? Or was this an ad hoc visual observations made by the authors?*

The anterior-posterior ordering of the SNPs and corresponding cortical regions was an ad hoc visual observation made when plotting results. However, we now also provide a supplementary table with the anterior-posterior ordering of the SNPs according to their position on chromosome 17, and corresponding brain regions alongside a rank correlation in Supplementary Table 23. However, we are cautious to not over-emphasize this finding as such gradient analyses usually involve a much larger number of regions to ensure stability of the correlation. We had no prior expectation of this pattern based on literature when performing the analysis.

• *Results (general): Conditional analyses, controlling for the most-significant variant(s), could be helpful in reinforcing some of the statements about multiple signals in a locus (beyond simply having LD below the fairly liberal threshold of $r^2 < 0.4$).*

In response to the Reviewer’s suggestion, we have updated the comparative analyses between ENIGMA data and our findings in the UKB by providing an exact LD value for each pair of risk loci that we identified with the highest LD SNP from the ENIGMA analyses of global and regional cortical thickness phenotypes in Supplementary Table 2. This addition disambiguates the strength of association between the genetic risk loci found in ENIGMA and the genetic risk loci we identified in the UKB. It’s a ‘trans’ comparison in that we tested the LD between each of our genetic risk loci and the genetic risk loci for all cortical thickness phenotype in ENIGMA and reported the highest phenotype from ENIGMA.

The Reviewer raises another important point regarding the potential for multiple independent signals at the same locus. In particular, a challenge in analyses of cortical thickness is that we have 34 cortical thickness phenotypes and a number of genetic variants from the same genomic region that are associated with these phenotypes. We have therefore conducted a multivariate regression with partial least squares (PLS) on the subregion of chromosome 17 to test whether a latent variables approach could capture independent signals not only within this region but also across all cortical thickness regions simultaneously. Based on this analysis, we have added the following to our main text Results and Supplementary Information:

“Results:

Using partial least squares regression (Supplementary Section 7), we also show that while some SNPs were uniquely linked to cortical thickness of specific regions, there were also multivariate patterns of association as several SNPs were linked to the cortical thickness of several regions.”

“Supplementary Section 7. Partial least squares regression analysis of a subregion of chromosome 17

We used a partial least squares regression with 12 genetic risk loci SNPs from a subregion of chromosome 17 visualized in Figure 3 as predictors (34,660x12) and 13 cortical thickness variables as outcomes (34,660x13). Both the SNPs and the cortical regions were selected based on the GWAS results shown in Supplementary Table 2. We regressed out age, age x sex, age², sex and TIV from the cortical thickness data and age, age x sex, age², sex, and the first 10 genetic PCs from the SNP data before entering the residuals into the PLS regression. We found that the overall PLS with 12 latent components model explained significantly more variance in cortical thickness than expected by chance^{1,20} (Supplementary Figure 10A). We show the SNP loadings for the 12 latent variables in Supplementary Figure 10B and the correlations

between latent variables and the cortical thickness data in Supplementary Figure 10C. While the first few latent variables captured more general patterns of association featuring multiple regions and genetic variants, we also found several variants with more specific associations with regional cortical thickness.”

“Supplementary Figure 10. Multivariate partial least squares (PLS) regression of genetic risk loci on chromosome 17 (located between 43,009,797 and 44,353,728) and cortical thickness phenotypes. The PLS model explained significantly more variance in cortical thickness than expected by chance (A), with predicted vs observed correlations for the 13 cortical thickness phenotypes ranging between 0.038 and 0.059 (e.g. D). Panel B shows the PLS loadings linking individual SNPs to the latent variables (LVs), with higher loadings indicating a greater contribution to the LVs. Panel C shows the correlations between latent variable scores and each cortical thickness (* $p_{FDR}<0.1$). a.u.: arbitrary units.

• Results-GxE: “combing” -> “combining”

We thank the Reviewer for pointing this out and have corrected this in text.

• Fig. 4: Panels E,F may not be necessary – without genomic context or other annotations, this visual information doesn’t seem to add much compared to a simple table.

Following the Reviewer’s feedback, we have updated Figure 4 to remove the panels E and F showing individual SNPs and instead present the total number of significant SNPs in the figure and the details on the SNPs in the Supplementary Tables.

• Results-GxE: How were the attributable proportions of variance explained by G & GxE (for example, for rs11126806) calculated?

To calculate proportions of variance explained, we leveraged the model fit statistic from the linear model package (fitlm, MATLAB R2016b), which provides ordinary r^2 values for several models: first, a base model includes the full set of covariates minus the main effects of the SNP, second, the main effect model includes the main effects of the SNP; and third, the interaction model that also adds the SNP x cardiovascular/depression interaction. Attributable variance was calculated as the difference in model R2 between term inclusive vs. exclusive models. We have clarified this in the main text:

“For example, compared to the base model including all covariates, the addition of the main effects of rs11126806 explained 0.08% of variance in regional thickness, while the addition of the interaction explained an extra 0.08% of variance in regional thickness compared to the main effect model.”

• *Results-GxE: Am I correctly understanding that the ROS/MAP replication of SNP-specific GxEs was achieving p-values <0.0021 even with only 66 samples? Where did that p-value threshold come from?*

Due to the low power for replication in ROS/MAP, we have replaced this analysis with a cross-validation of our results in UKB. The ROS/MAP replication sample included 202 participants with SNP data (66 samples included the post mortem gene expression data); we were indeed achieving p-values of <0.0021; we used the same approach in ROS/MAP as in UKB, with the regions showing an interaction being somewhat different in ROS/MAP compared to UKB. Following a suggestion by Reviewer 1, we instead include a cross-validation analysis of the UKB data in the Supplementary:

“Supplementary Section 2. Cross-validation analysis of GxE interactions

In order to test the robustness of our GxE findings, we repeated all analyses using cross-validation. For each of the 220 candidate SNPs, we first split our sample in 10 folds. For each split, we fit linear models in 90% of the sample, creating a GxE interaction brain map (featuring 360 regions) for depression and cardiovascular disease. We then thresholded each brain map at uncorrected $p < 0.01$ in the inner fold (90% of the sample), and proceeded to test for interactions in the outer fold, i.e. the held-out 10% of the data. For regions that passed the $p < 0.01$ threshold in the inner fold, we averaged the cortical thickness values in the outer fold and obtained an interaction p-value. Where no significant regions were identified, we used global cortical thickness as an outcome. We repeated the process 10 times, testing each combination of held-out outer fold and inner fold data. This cross-validation procedure allowed us to test the robustness of the regions with an interaction in held-out data. As a result, for each SNP, we obtained 10 p-values for each of the cross-validation splits. We combined the p-values using Fisher’s method⁹. An overview of the approach is shown in Supplementary Figure 3.

The summary statistics for whole-sample analyses and cross-validation analyses for cardiovascular disease and depression are shown in Supplementary Tables 21 and 22, respectively. We found that many of the SNPs that passed the stringent significance threshold in the main analyses ($p < 0.05/220/18.1$ effective comparisons) also passed the uncorrected $p < 0.05$ threshold in the cross-validation analyses. We found that 20 of the 60 SNPs with a significant interaction with cardiovascular disease in the main analyses were also significant at uncorrected $p < 0.05$ following cross-validation. Among 46 SNPs with a significant interaction with depression in the main analyses, 7 SNPs were also significant at uncorrected $p < 0.05$ following cross-validation.”

• *Results-GxE: It would be helpful to have clarification or citations related to the use of the terms “multiplicative” and “additive”. It seems like the authors are using these terms to describe what are often referred to as “qualitative” (changing effect signs) and “quantitative” (changing only magnitudes) interactions. The terms “multiplicative” and “additive” are usually reserved for distinct types of statistical tests, often in the context of binary outcomes.*

We thank the Reviewer for this clarification – following the Reviewer’s suggestion, we have added the below reference that provides more detail on the different naming conventions that feature both ‘qualitative/additive’ and ‘multiplicative/quantitative’ nomenclature:

“Most interactions were multiplicative quantitative in nature²¹, with opposite effects of the SNP observed in cases compared to controls, although some additive interactions with the same effect direction, but different magnitude in cases and controls were also found (Supplementary Figure 4).”

“21. T. J. Van Der Weele, M. J. Knol, A tutorial on interaction. *Epidemiol. Method.* **3**, 33–72 (2014).”

We have also amended Figure 4 to show that for the three example SNPs plotted, the large effects of disease condition were modified by the genotype category.

REVIEWER COMMENTS

Reviewer #1 (Remarks to the Author):

- The reviewer maintains reservations regarding the paper's assertion that "Influences of modifiable risk factors for dementia and gene-by-environment interactions" This highlights the potential bias caused by the study's unclear definition of risk factors, significantly impacting result interpretation. This stands as the reviewer's primary concern. Notably, within the UKBB, only a small fraction of participants was diagnosed as Alzheimer's disease (AD) ($N < 10$ for the imaging population), and definitions of depression and cardiovascular factors vary widely and are very heterogeneous. The current criteria included by the authors in their current analyses pose a potential bias that affects all subsequent analyses.
- The authors state: "Among the 22 loci significantly associated with global thickness, only five were in high LD ($r^2 > 0.4$) with the genome-wide significant loci from the most recent ENIGMA GWAS 24 , likely since we did not include global thickness as a covariate and due to the more homogeneous the makeup of our UKB sample compared to ENIGMA.". This doesn't seem to be true. The ENIGMA Science paper indeed included the UKBB participants. I am surprised that the overlaps here are quite low - only 5 of 22 were in LD or replicated.
- The authors should justify their choice to use PLINK for the GWAS. More advanced linear mixed effect models can boost the statistical power compared to PLINK.
- How does the author reconcile the difference in genetics sequencing between the UKBB and CLSA data? I have not seen details of the genetic QC for CLSA genetic data. Is it genotype, imputed data, or WGS?
- The authors performed many analyses using different resources and datasets, but the rationale behind them is unclear. For example, the authors tested the gene ontology analysis, but FUMA's Gene2Func provides this functionality directly.
- SNP-to-Gene mapping was done in SNP2GENE, not Gene2Func in FUMA: "For annotation

of genomic risk loci in each GWAS, we used FUMA's Gene2Func tool 44 ."

- No details were given regarding how the PRS was calculated. Which method did the author use? What are the QC steps included? Overall, details of the methodologies used in the study are not fully reproducible for readers.

Overall, this strong paper presents large-scale analyses by combining multiple data resources. However, the main results do not fully support the title's main message – the main advance compared to previous cortical thickness GWAS.

Reviewer #2 (Remarks to the Author):

I commend the authors on their updates to the manuscript and comprehensive responses addressing prior comments. In general, the manuscript is nicely improved. Most of my concerns are addressed, though I'll add a few additional notes below:

- I appreciate the updates to the Title and Intro reflecting the fact that much of this investigation is about genetic main effects, rather than the entire paper being focused on interactions. However, the abstract could still use a similar update. Right now, the objective and most results presented in the abstract are focused on the interaction question only, despite the fact that 4/6 of the Results sections are not related to interactions.

- I'm still not sure, based on the authors' description of the FUMA GWAS-to-eQTL linking strategy, that this can be called "fine mapping". It might be more appropriate to just directly call this eQTL mapping, or variant-to-transcript mapping, or something like that. My concern is only the "fine mapping" has a specific connotation within the genetic epidemiology field related to assigning probabilities that a given variant or group of variants within a locus is causal.

- The use of PLS to incorporate multiple genetic variants within a locus is an interesting approach. It might be helpful for the authors to make a note about this choice compared to a formal haplotype analysis, a traditional approach to dealing with multiple linked variants in a locus.

- I don't agree with the changes that the authors made regarding interaction nomenclature

(additive/multiplicative and quantitative/qualitative). SNPs with opposite directions of effect in cases vs controls (or any two strata) indicate a qualitative, not quantitative, interaction. Likewise, situations with the same effect direction but different magnitude are typically described as quantitative (not additive) interactions. The additive/multiplicate distinction is indeed discussed in the VanderWeele and Knol paper, but it is a separate concept than the one the authors are invoking.

Below we respond to each of the Reviewer's comments (Reviewer feedback is presented in black, while our responses are shown in blue. Quoted text from the manuscript or supplementary is shown in blue, with added text underlined).

REVIEWER COMMENTS

Reviewer #1 (Remarks to the Author):

• The reviewer maintains reservations regarding the paper's assertion that "Influences of modifiable risk factors for dementia and gene-by-environment interactions" This highlights the potential bias caused by the study's unclear definition of risk factors, significantly impacting result interpretation. This stands as the reviewer's primary concern. Notably, within the UKBB, only a small fraction of participants was diagnosed as Alzheimer's disease (AD) (N<10 for the imaging population), and definitions of depression and cardiovascular factors vary widely and are very heterogeneous. The current criteria included by the authors in their current analyses pose a potential bias that affects all subsequent analyses.

We agree with the Reviewer that there are varying definitions of cardiovascular disease factors and depression. We used self-report, because such an approach, despite its limitations, offers a more inclusive definition of risk factors for dementia. In the manuscript, we state the specific data fields used to define these conditions, such that it is clear for readers. Self-report definitions, while less specific than ICD-based definitions, have been previously used for various conditions including depression (Cai et al Nature Genetics 2020 <https://www.nature.com/articles/s41588-020-0594-5>). It is well-known that cardiovascular disease and depression are risk factors for AD, and that brain changes indicative of incipient AD appear many years prior to formal diagnosis. Our approach was less about people with AD, than about studying risk factors related to AD in mid to late life, before AD onset. As such, we investigated the effects of cardiovascular and depression risk factors for AD on proxies of brain health, i.e. cortical thickness and white matter hyperintensities, which is an approach that has been used many times to better understand biomarkers of AD risk (e.g. Schwarz, C. G. et al. A large-scale comparison of cortical thickness and volume methods for measuring Alzheimer's disease severity. *NeuroImage Clin.* 11, 802–812 (2016) and Sperling, R. A. et al. Toward defining the preclinical stages of Alzheimer's disease: Recommendations from the National Institute on Aging- Alzheimer's Association workgroups on diagnostic guidelines for Alzheimer's disease. *Alzheimer's Dement.* 7, 280–292 (2011)), and that has been validated in the sense that early changes in these markers increase the likelihood of AD diagnosis in later-life. However, we agree with the Reviewer that while cardiovascular conditions and depression are risk factors for AD, they also increase the risk for other medical conditions and have amended the title as follows to remove the bias towards dementia:

“Genetic influences on brain and cognitive health and their interactions with cardiovascular conditions and depression”

We have also added the following section to the Discussion to acknowledge that we are not studying people with dementia, but rather focusing on markers of structural brain health (added text underlined):

“The UKB sample is ideal to assess the GxE effects given its size and the age range (45-81) of its participants. In this population sample there are very few people with dementia; however, it is possible to study biomarkers of preclinical AD by leveraging neuroimaging measures of brain health. In particular, cortical thinning is a good proxy measure of ADRD severity¹⁵, including incipient dementia^{77,78}.”

We have also amended the abstract in relation to the Reviewer's comment. In particular, we aim to achieve more balance between our GxE findings and the main effects of genetic risk loci on brain and cognitive health:

“Approximately 40% of dementia cases could be prevented or delayed by modifiable risk factors related to lifestyle and environment. These risk factors, such as depression and vascular disease, do not affect all individuals in the same way, likely due to inter-individual differences in genetics. However, the precise nature of how genetic risk profiles interact with modifiable risk factors to affect brain health is poorly understood. Here we combine multiple data resources, including genotyping and postmortem gene

expression, to map the genetic landscape of brain structure and identify 367 loci associated with cortical thickness and 13 loci associated with white matter hyperintensities ($P < 5 \times 10^{-8}$). We show that among 220 unique genetic loci associated with cortical thickness in our genome-wide association studies (GWAS), 95 also showed evidence of interaction with depression or cardiovascular conditions. Polygenic risk scores based on our GWAS of inferior frontal thickness also interacted with hypertension in predicting executive function in the Canadian Longitudinal Study on Aging. These findings advance our understanding of the genetic underpinning of brain structure. ~~They and show that genetic risk for brain and cognitive health is in part moderated by treatable mid-life factors, which represents targets for early interventions.~~

• The authors state: "Among the 22 loci significantly associated with global thickness, only five were in high LD ($r^2 > 0.4$) with the genome-wide significant loci from the most recent ENIGMA GWAS 24, likely since we did not include global thickness as a covariate and due to the more homogeneous the makeup of our UKB sample compared to ENIGMA.". This doesn't seem to be true. The ENIGMA Science paper indeed included the UKBB participants. I am surprised that the overlaps here are quite low - only 5 of 22 were in LD or replicated.

While the Grasby et al. analysis included 10,347 UKB participants (28% of the sample), it also included 26,589 participants from other studies, such as ENIGMA and ADNI (as listed in Grasby et al. Supplementary Table 2). Among the non-UKB participants in Grasby's study, there were many small subsamples coming from different sites with different psychiatric conditions such as schizophrenia, depression and Alzheimer's/MCI ($n=9624$, i.e 26%), and 16,965 (46%) participants from other population-based studies. We reproduce a subset of the supplementary table 2 from Grasby et al to provide a more detailed breakdown of the samples in the ENIGMA GWAS in the table below. Therefore, we feel it is sensible to state that the makeup of the overall ENIGMA GWAS sample was quite different from the UKB samples studied in our paper. Further, as stated in the results section cited by the Reviewer, we compare the ENIGMA GWAS results that co-varied for cortical thickness with our findings that did not co-vary for global thickness. We therefore reason that these two factors (co-varying for global thickness and the heterogeneous samples in Grasby et al.) could account for the observed differences. Despite the lower overlap in genomic risk loci, we do find a high genetic correlation between our summary statistics and those reported by Grasby et al. (as reported in our original submission):

"Compared to published GWAS of thickness and WMH, we observed the expected strong genetic correlations between our results and those from the ENIGMA consortium's global thickness analysis ($r_g=0.82$, $p=5 \times 10^{-95}$) and from Persyn et al.'s analysis of WMH²⁷ ($r_g=0.976$, $p=6 \times 10^{-70}$)."

We therefore believe that the observed level of overlap between our findings and the Grasby/ENIGMA findings is in line with expectations.

[table redacted]

Table 1. An overview of discovery samples from Grasby et al. 2020, adapted from Grasby et al. Supplementary Table 2, sorted by study type.

• The authors should justify their choice to use PLINK for the GWAS. More advanced linear mixed effect models can boost the statistical power compared to PLINK.

We appreciate this comment and provide a justification for using PLINK below. We chose PLINK for several reasons: 1) PLINK is particularly well-suited to the analysis of large datasets given its efficiency and integration with auxiliary tools and methods^{a,b,c}, 2) PLINK is a well-established and commonly used software for GWAS analyses and was used by Grasby et al. and Hofer et al. in their previously published cortical thickness GWASs. We agree that LMM-based approaches offer some benefits when cryptic relatedness and other sources of population substructure are present, though independent analyses have shown that these gains may be less pronounced with SNPs crossing genome-wide levels of significance

(5.0×10^{-8}) <https://www.ncbi.nlm.nih.gov/pmc/articles/PMC6980752/> and come at the cost of computational efficiency (which was meaningful given the large number of GWAS performed in our study). We meta-analyzed the PLINK results from White British European and non-White British European participants using METAL to improve the statistical power of our analyses and avoid this potential source of stratification. We have added these justifications, and accompanying limitations of the PLINK-based additive OLS modeling approach, to our discussion:

“Our study has several limitations. First, the UKB includes participants with a predominantly European ancestry. Larger well-phenotyped non-Caucasian samples are needed for our results to generalize across ancestries^{87–89}. While PLINK-based linear modeling is computationally efficient, it may result in inflated type I error rates when population substructure is present in the study sample; future studies including more diverse populations would benefit from emerging mixed-effect models designed for biobank-scale analyses^{90,91}.”

90. Mbatchou, J. *et al.* Computationally efficient whole-genome regression for quantitative and binary traits. *Nat. Genet.* **53**, 1097–1103 (2021).

91. Jiang, L., Zheng, Z., Fang, H. & Yang, J. A generalized linear mixed model association tool for biobank-scale data. *Nat. Genet.* **53**, 1616–1621 (2021).

a. Marees, A.T., de Kluiver, H., Stringer, S., Vorspan, F., Curis, E., Marie-Claire, C. and Derks, E.M., 2018. A tutorial on conducting genome-wide association studies: Quality control and statistical analysis. *International journal of methods in psychiatric research*, 27(2),

b. Purcell, S., Neale, B., Todd-Brown, K., Thomas, L., Ferreira, M.A., Bender, D., Maller, J., Sklar, P., De Bakker, P.I., Daly, M.J., Sham, P.C. 2007. PLINK: a tool set for whole-genome association and population-based linkage analyses. *The American journal of human genetics*, 81(3), pp.559-575.

c. Uffelmann, E., Huang, Q.Q., Munung, N.S., De Vries, J., Okada, Y., Martin, A.R., Martin, H.C., Lappalainen, T. and Posthuma, D. 2021. Genome-wide association studies. *Nature Reviews Methods Primers*, 1(1)

• How does the author reconcile the difference in genetics sequencing between the UKBB and CLSA data? I have not seen details of the genetic QC for CLSA genetic data. Is it genotype, imputed data, or WGS?

We thank the Reviewer for bringing this to our attention and agree that more information on CLSA genetic data processing is needed. We have therefore added the following sentence to the Methods to more explicitly provide the information on genotype data in CLSA. Reassuringly, the CLSA paper describing the genotype data states that the CLSA QC procedures followed UK Biobank documentation:

“CLSA genetic data processing, including genotype QC and imputation to the TOPMed reference panel is described in more detail in previous studies³¹. CLSA QC procedures followed UK Biobank QC documentation³¹.”

“31. V. Forgetta, R. Li, C. Darmond-Zwaig, A. Belisle, C. Balion, D. Roshandel, C. Wolfson, G. Lettre, G. Pare, A. D. Paterson, L. E. Griffith, C. Verschoor, M. Lathrop, S. Kirkland, P. Raina, J. B. Richards, J. Ragoussis, Cohort profile: Genomic data for 26 622 individuals from the Canadian Longitudinal Study on Aging (CLSA). *BMJ Open* **12** (2022), doi:10.1136/bmjopen-2021-059021.”

• The authors performed many analyses using different resources and datasets, but the rationale behind them is unclear. For example, the authors tested the gene ontology analysis, but FUMA's Gene2Func provides this functionality directly.

We thank the Reviewer for bringing this to our attention. While we considered using Gene2Func, ClusterProfiler GO analysis allows for more flexible visualization of results: while Gene2Func works on one outcome/phenotype at a time, ClusterProfiler can visualize a matrix of phenotypes x GO terms as in the Supplementary Figure 5. In order to clarify the choices of methods and datasets within the narrative of our study, we have added several new explanatory statements to the methods (underlined), as each analysis is introduced:

“Main effect GWAS models

We used PLINK2 (<https://www.cog-genomics.org/plink/2.0/assoc>)⁴¹ to independently test for additive allelic dosage associations with a) WMH, b) global cortical thickness calculated as the average thickness of the cortex, and c) 33 regional thickness in the large-scale UKB data.

...

To replicate our findings, we tested for additive associations between 220 SNPs identified as genetic risk loci for cortical thickness in UKB analyses in ROS/MAP data.”

“... Finally, to identify biological processes implicated by GWAS analyses of regional cortical thickness, we used the *clusterProfiler* package in R (4.2.0) to conduct gene ontology (GO) overrepresentation analyses on position-mapped (ANNOVAR) genes, which maps several GO terms to several phenotypes in the same figure.”

“Postmortem brain differential expression analysis

Postmortem bulk tissue RNAseq data for DLPFC from participants in ROS/MAP were accessed to identify genes whose expression was associated with cortical thickness.”

“Gene effects on cognitive function

Using cognitive and genetic data from the CLSA, we aimed to test whether genetic risk loci for cortical thickness also affect cognitive function and whether polygenic scores for cortical thickness interact with hypertension to affect cognitive function. We first tested for associations of each variant from UKB GWAS analyses with composite scores of memory and executive function^{29,30}.”

- SNP-to-Gene mapping was done in SNP2GENE, not Gene2Func in FUMA: "For annotation of genomic risk loci in each GWAS, we used FUMA's Gene2Func tool 44 ."

We thank the Reviewer for pointing out this typographical mistake, and have corrected the sentence as below:

“For annotation of genomic risk loci in each GWAS, we used FUMA's SNP2GENE ~~Gene2Func~~ tool⁴⁵.”

- No details were given regarding how the PRS was calculated. Which method did the author use? What are the QC steps included? Overall, details of the methodologies used in the study are not fully reproducible for readers.

In calculating the PRS, we multiplied the aligned effect allele dosage with the METAL summary statistics from the GWAS and added together the resulting values – a standard additive approach often named “clumping and thresholding” – as described:

“Second, we calculated a polygenic score in CLSA for each brain region by multiplying the aligned effect allele dosage (0,1,2) with the summary statistics from the GWAS analysis (Z-score) and adding them together to obtain the polygenic score.”

To ensure transparency and reproducibility, the code for performing our PRS calculations is available at https://github.com/peterzhukovsky/imaging_genetics/blob/main/GxE/CLSA/CLSA_PRS.m and shared as part of our submission (link included in our data availability statement).

We have also provided more information regarding the QC and imputation of underlying CLSA data that was used to calculate our PRSs in response to the Reviewer's question above.

Overall, this strong paper presents large-scale analyses by combining multiple data resources. However, the main results do not fully support the title's main message – the main advance compared to previous cortical thickness GWAS.

We thank the Reviewer for their constructive and encouraging feedback, and have updated the title and abstract to better represent our study's conclusions and unique contributions.

Reviewer #2 (Remarks to the Author):

I commend the authors on their updates to the manuscript and comprehensive responses addressing prior comments. In general, the manuscript is nicely improved. Most of my concerns are addressed, though I'll add a few additional notes below:

- I appreciate the updates to the Title and Intro reflecting the fact that much of this investigation is about genetic main effects, rather than the entire paper being focused on interactions. However, the abstract could still use a similar update. Right now, the objective and most results presented in the abstract are focused on the interaction question only, despite the fact that 4/6 of the Results sections are not related to interactions.

We thank the Reviewer for their helpful feedback and have further updated the title and abstract. We specifically refer to 'cardiovascular conditions and depression' instead of 'risk factors for dementia' following a similar point raised by Reviewer 1:

New title: "Genetic influences on brain and cognitive health and their interactions with cardiovascular conditions and depression"

Old title: "Genetic influences on brain and cognitive health and their interactions with modifiable risk factors for dementia"

In addition, we have updated the abstract as follows (added text underlined):

"Approximately 40% of dementia cases could be prevented or delayed by modifiable risk factors related to lifestyle and environment. These risk factors, such as depression and vascular disease, do not affect all individuals in the same way, likely due to inter-individual differences in genetics. However, the precise nature of how genetic risk profiles interact with modifiable risk factors to affect brain health is poorly understood. Here we combine multiple data resources, including genotyping and *post mortem* gene expression to map the genetic landscape of brain structure and identify 367 loci associated with cortical thickness and 13 loci associated with white matter hyperintensities ($P < 5 \times 10^{-8}$). We show that among 220 unique genetic loci associated with cortical thickness in our genome-wide association studies (GWAS), 95 also showed evidence of interaction with depression or cardiovascular conditions. Polygenic risk scores based on our GWAS of inferior frontal thickness also interacted with hypertension in predicting executive function in the Canadian Longitudinal Study on Aging. These findings advance our understanding of the genetic underpinning of brain structure. They and show that genetic risk for brain and cognitive health is in part moderated by treatable mid-life factors, which represents targets for early interventions."

- I'm still not sure, based on the authors' description of the FUMA GWAS-to-eQTL linking strategy, that this can be called "fine mapping". It might be more appropriate to just directly call this eQTL mapping, or

variant-to-transcript mapping, or something like that. My concern is only the "fine mapping" has a specific connotation within the genetic epidemiology field related to assigning probabilities that a given variant or group of variants within a locus is causal.

We agree with the Reviewer that eQTL mapping is a more accurate description for the SNP2GENE mapping using eQTL in FUMA and have removed the reference to fine mapping as below, ensuring all other references to the SNP2GENE analyses include eQTL mapping throughout the manuscript:

"In parallel, we performed ~~computational fine mapping~~ eQTL mapping of genetic loci from our GWAS using eQTL data from GTEx (using frontal cortex and BA9 references in FUMA)."

- The use of PLS to incorporate multiple genetic variants within a locus is an interesting approach. It might be helpful for the authors to make a note about this choice compared to a formal haplotype analysis, a traditional approach to dealing with multiple linked variants in a locus.

We thank the Reviewer for this suggestion and have updated the results and supplementary sections to include an improved explanation and discussion of our PLS analysis:

“RESULTS

To disentangle the associations between multiple genetic variants and multiple phenotypes, we used partial least squares regression (Supplementary Section 7). We show that while some SNPs were uniquely linked to cortical thickness of specific regions, we also found multivariate patterns of association, as several SNPs were linked to the cortical thickness of several regions.”

“Supplementary Section 7

We used a partial least squares regression with 12 genetic risk loci SNPs from a subregion of chromosome 17 visualized in Figure 3 as predictors (34,660x12) and 13 cortical thickness variables as outcomes (34,660x13). We chose to apply this multivariate latent model instead of a more traditional haplotype analysis given that a) haplotype construction from our discovered SNPs within such a large region with complex patterns of LD, including several unlinked variants, would have yielded many haplotypes with very low frequencies and b) we wanted to assess the relationships between all variants and several imaging phenotypes simultaneously.”

We had also tested for associations of the 12 genetic risk loci from the chromosome 17 and haplotype-tagging SNPs for the *MAPT* haplotype (Supplementary Figure S6) in the original submission:

“Results

Given the presence of a large *MAPT* haplotype at 17q21⁶², we also report the associations of haplotype tagging SNPs⁶³ with global cortical thickness and LD between these SNPs with loci we show in Figure 3A. We found signals for cortical thickness both within and outside of the *MAPT* haplotype. Briefly, a tagging SNP for the protective H2 haplotype (rs8070723) was in high LD with several SNPs identified in our GWAS (Supplementary Figure 6) and was associated with global ($p=4\times 10^{-6}$) and regional thickness (Supplementary Table 19). However, our GWAS also identified other SNPs from this region that were not in LD with the *MAPT* haplotype.”

“Supplementary Figure 6. Linkage disequilibrium plot between SNPs risk loci associated with cortical thickness in 17q21 (specifically the 43,009,797-44,353,728 region) and haplotype-tagging SNPs (htSNPs) for the *MAPT* gene. Associations with global cortical thickness are shown in red ($p<5\times 10^{-8}$), orange ($p<5\times 10^{-5}$) and gray ($p>0.001$) to

denote significance. LD values shown are r^2 , multiplied by 100.”

- I don't agree with the changes that the authors made regarding interaction nomenclature (additive/multiplicative and quantitative/qualitative). SNPs with opposite directions of effect in cases vs controls (or any two strata) indicate a qualitative, not quantitative, interaction. Likewise, situations with the same effect direction but different magnitude are typically described as quantitative (not additive) interactions. The additive/multiplicative distinction is indeed discussed in the VanderWeele and Knol paper, but it is a separate concept than the one the authors are invoking.

We thank the Reviewer for this suggestion and have amended our language throughout the manuscript around the interpretation of the interactions to focus on qualitative and quantitative interactions (added changes underlined) instead of ‘multiplicative’ and ‘additive’:

“Most interactions were qualitative in nature⁶⁴, with opposite effects of the SNP observed in cases compared to controls, although some additive quantitative interactions with the same effect direction but different effect magnitude in cases vs. controls were also found (Supplementary Figure 4).”

We had previously referred to interactions as multiplicative and additive in the supplementary, but now we have also amended Supplementary Figure 4 as below:

“Supplementary Figure 4. Overview of interaction types, showing T-statistics for the main effect of SNPs on regional cortical thickness.

SNP main effect on SNP-specific cortical thickness in all participants without cardiovascular problems (T-statistic)

SNP main effect on SNP-specific cortical thickness in all participants without depression (T-statistic)

REVIEWERS' COMMENTS

Reviewer #1 (Remarks to the Author):

This study is an extensive study in a rapidly growing field. The authors have provided an extensive response to comments, albeit this reviewer continues to have some reservations. For example, the PLINK linear model has been outdated. A Science paper published 4 years ago using PLINK might not fully support such modeling choices. The authors also seem to have used several terms in unusual ways, e.g. fine-mapping vs. SNP annotations and FUMA functionalities. My main reservation is consistent with my previous comments - it is not clearly convincing right now in what ways does this study really advances the field from previous literature (GWAS on cortical thickness) or that it properly conveys the main advancements in the title. Such advancements could include statistical methods, data, or features, especially since Reviewer #2 also commented that this study's primary results are on the genetic association, not the interactions or gene-environmental interactions. Prior studies of relevance:

<https://www.science.org/doi/10.1126/sciadv.abj9446>

<https://www.science.org/doi/10.1126/science.aay6690>

Below we respond to each of the Reviewer's comments (Reviewer feedback is presented in black, while our responses are shown in blue. Quoted text from the manuscript or supplementary is shown in blue, with added text underlined).

REVIEWER COMMENTS

Reviewer #1 (Remarks to the Author):

This study is an extensive study in a rapidly growing field. The authors have provided an extensive response to comments, albeit this reviewer continues to have some reservations. For example, the PLINK linear model has been outdated. A Science paper published 4 years ago using PLINK might not fully support such modeling choices.

While we agree with the Reviewer that mixed effect models such as REGENIE may be preferable under certain conditions (e.g. in datasets with greater ancestral diversity), we believe the use of PLINK2 in the UKB sample coupled with METAL meta-analysis is an acceptable approach for our use case. We have added a reference to the potential use of multivariate and mixed-effect models in future work in our limitations section of the discussion:

“While PLINK-based linear modeling is computationally efficient, it may result in inflated type I error rates when population substructure is present in the study sample; future studies including more diverse populations would benefit from emerging multivariate¹⁸ and mixed-effect models designed for biobank-scale analyses^{65,66}.”

18. Van Der Meer, D. *et al.* The genetic architecture of human cortical folding. *Sci. Adv.* **7**, 1–10 (2021).
65. Mbatchou, J. *et al.* Computationally efficient whole-genome regression for quantitative and binary traits. *Nat. Genet.* **53**, 1097–1103 (2021).
66. Jiang, L., Zheng, Z., Fang, H. & Yang, J. A generalized linear mixed model association tool for biobank-scale data. *Nat. Genet.* **53**, 1616–1621 (2021).

The authors also seem to have used several terms in unusual ways, e.g. fine-mapping vs. SNP annotations and FUMA functionalities.

We have updated the terminology following feedback from both Reviewers.

My main reservation is consistent with my previous comments - it is not clearly convincing right now in what ways does this study really advances the field from previous literature (GWAS on cortical thickness) or that it properly conveys the main advancements in the title. Such advancements could include statistical methods, data, or features, especially since Reviewer #2 also commented that this study's primary results are on the genetic association, not the interactions or gene-environmental interactions.

Overall, we have de-emphasized the importance of the GxE interactions in the abstract (by adding more focus on main effects) and in the title (in the previous revision). We believe the previous revision of the title was sufficient; we have also emphasized the main effects and the genetic associations with cognition in the abstract.

Prior studies of relevance:

<https://www.science.org/doi/10.1126/sciadv.abj9446>
<https://www.science.org/doi/10.1126/science.aay6690>

We thank the reviewer for bringing these studies to our attention and cite them throughout the manuscript.